# Adaptive higher order reversible integrators for memory efficient deep learning

## Abstract

The depth of networks plays a crucial role in the effectiveness of deep learning. However, the memory requirement for backpropagation scales linearly with the number of layers, which leads to memory bottlenecks during training. Moreover, deep networks are often unable to handle time-series data appearing at irregular intervals. These issues can be resolved by considering continuous-depth networks based on the neural ODE framework in combination with reversible integration methods that allow for variable time-steps. Reversibility of the method ensures that the memory requirement for training is independent of network depth, while variable time-steps are required for assimilating time-series data on irregular intervals. However, at present, there are no known higher-order reversible methods with this property. High-order methods are especially important when a high level of accuracy in learning is required or when small time-steps are necessary due to large errors in time integration of neural ODEs, for instance in context of complex dynamical systems such as Kepler systems and molecular dynamics. The requirement of small time-steps when using a low-order method can significantly increase the computational cost of training as well as inference. In this work, we present an approach for constructing high-order reversible methods that allow adaptive time-stepping. Our numerical tests show the advantages in computational speed when applied to the task of learning dynamical systems.

## 1 Introduction

Deep neural networks are widely used across various learning tasks (Russakovsky et al., 2015; Esteva et al., 2017), and their depth often plays a crucial role in the effectiveness of learning. These networks have also been shown to be particularly useful in the tasks of learning models of dynamical systems (Chen et al., 2018; Raissi et al., 2018a; Rudy et al., 2019b; Schüssler et al., 2019; Rudy et al., 2019a; Liu et al., 2022; Raissi et al., 2018b). It was recently shown that the use of numerical methods for neural network architectures can provide impressive results with theoretical guarantees (Haber & Ruthotto, 2017; Chang et al., 2018; Celledoni et al., 2021; Maslovskaya & Ober-Blöbaum, 2024). In this work we use the theory of symmetric numerical methods for the construction of a new class of reversible neural networks. The new class allows memory efficient computations of gradients in training and reduced computational costs in learning models of dynamical systems, where the parameters typically need to be identified to high accuracy and the depth of the networks can be very large. Our network architecture constitutes an important contribution to ensure scalability of neural ODEs to high-dimensional dynamical systems that arise, for instance, as discretizations of systems governed by partial differential equations.

The high memory costs of computing the gradient of very deep neural networks using the backpropagation algorithms poses a significant bottleneck in their training, hindering their scalability and efficiency. To address this, a neural ODE approach combined with the adjoint method has been proposed for gradient computations (Chen et al., 2018), which avoids storing intermediate states during forward propagation, potentially making the cost of gradient computation independent of network depth. However, it was quickly realized (Gholaminejad et al.; Zhuang et al., 2020) that using this approach with arbitrary discretization methods leads to incorrect gradients.

The solution of this problem is to use reversible integrators, which ensure accurate gradient computations by reconstructing intermediate states precisely during backward integration. However, symplectic reversible integrators (Chang et al., 2018), which are a large class of well studied integra-

tors, do not allow adaptivity in the step size and require a particular structure in the neural ODE. This makes them unsuitable for use in time-series applications where data appears at irregular intervals, when time-steps need to be decreased adaptively to achieve a prescribed accuracy in the learning of dynamical systems, or when the identified model is used to predict continuous trajectories.

Adaptive time-stepping for numerical integrators for differential equations is a well-established field in numerical analysis (Hairer et al., 2013; Deuflhard & Bornemann, 2002). In adaptive time-stepping, step sizes for an integration step are selected such that an estimate for the local error is below a given error tolerance. In this way, computational cost in numerical integration can be saved when large step sizes are sufficient to obtain accurate results, while step sizes are automatically decreased when required. The only reversible methods compatible with variable time-step selection without losing the reversibility property are asynchronous leapfrog (ALF) (Mutze, 2016), which is based on the classical Verlet method (also known as leapfrog method) (Verlet, 1967), and the reversible Heun method (Kidger et al., 2021).

ALF has been used to construct neural network architectures known as MALI networks (Zhuang et al., 2021). These methods are based on operating on an augmented space: a neural ODE in the original variable $z$ is extended to a larger space and replaced by a neural ODE in $(z, v)$. Both methods are known to be of order of accuracy $(2, 1)$ in $(z, v)$ (Mutze, 2016), which makes them computationally costly in learning tasks that require high accuracy in the integration of the neural ODE, in particular, in learning of dynamical systems. To be able to reach high accuracy, the lower order methods are forced to use small step sizes and, as a result, have higher computational costs. This phenomenon was highlighted in the examples in (Matsubara et al., 2021). Furthermore, we show in C.1 that in parameter identification tasks there is a direct relation between the order of a numerical integrator and the order of accuracy of identified parameters. Therefore, there is a need for higher order reversible methods, which we address in this paper.

Various machine learning techniques have emerged in the past decades for approximating models of dynamical systems (Ghadami & Epureanu, 2022). These include methods based on Gaussian processes (Bouvrie & Hamzi, 2017; Raissi & Karniadakis, 2018; Hamzi & Owhadi, 2021), sparse regression on libraries of basis functions (Brunton et al., 2016; Tran & Ward, 2017; Reinbold et al., 2021), and recurrent neural networks (RNNs) (ichi Funahashi & Nakamura, 1993; Bailer-Jones et al., 1998; Karniadakis et al., 2021). In particular, the neural ODE approach (Chen et al., 2018) identified an important connection between the RNN structure and the numerical methods available for integration of differential equations. This was generalized by universal differential equations in (Rackauckas et al., 2021) for different types of differential equations. Other generalizations include physics informed learning of dynamical systems (Greydanus et al., 2019; Cranmer et al., 2020b; Jin et al., 2020; Chen et al., 2020) and operator approximation (Chen & Chen, 1995; Lu et al., 2021; Lin et al., 2023; Boullé & Townsend, 2024). If the training data consists of time-series data that corresponds to a constant time-step, neural ODEs can be trained with a low order method and time-series can be predicted with high accuracy provided that the trained neural ODE is integrated with the same integrator that was used during training (Zhu et al., 2021; David & Méhats, 2023; Offen & Ober-Blöbaum, 2022; Ober-Blöbaum & Offen, 2023). However, in realistic examples, snapshots of trajectories with variable time-steps need to be processed (Raissi et al., 2018b; Rudy et al., 2019b; Liu et al., 2022). Moreover, a discretization-independent prediction of the system's evolution is often desired and learning of underlying differential equations requires high accuracy in the learned parameters, which requires high accuracy simulation.

We demonstrate that our approach can be applied to high-dimensional dynamical systems, including those arising from discretizations of partial differential equations. This is a highly active research area. Other approaches in this context include model order reduction based techniques and operator inference (see the review (Kramer et al., 2024) or e.g. (Sharma et al., 2024; 2023; Allen-Blanchette et al., 2020; Mason et al., 2022)), or structure-preserving approaches for discrete field theories (Qin, 2020; Offen & Ober-Blöbaum, 2024; Offen, 2024).

The main contribution of this paper is the development of a methodology to construct reversible neural networks based on higher order numerical methods. First, we prove that, in contrast to the analysis in (Mutze, 2016), ALF is of order 2 in both $(z, v)$ at even time-steps. Using this property and the theory of composition methods, we construct a class of reversible networks of any even order. A particular architecture based on 4th order networks is compared to the already known ALF method

on examples of learning dynamical systems. The comparison of the two methods with adaptive time-stepping shows that the proposed higher order method is computationally more efficient.

## 2 BACKGROUND

### 2.1 NEURAL ODE

Assume that an unknown function $\mathcal{F} : X \to Y$ is approximated by a neural network based on training data $\{x_i, y_i = \mathcal{F}(x_i)\}_{i=1}^n$. The neural ODE approach to the deep network design employs the idea of continuous-depth networks and their discretization by a numerical method. The continuous-depth network is defined as a flow of a neural ODE of the form

$$\dot{z}(t) = f(z(t), \theta(t)), \quad z(0) = (x_1, \dots, x_n), \tag{1}$$

on the time interval $[0, T]$ for some vector field $f$. Discretization of a neural ODE with a numerical method of step size $h$ is defined as follows

$$z_{j+1} = \sigma_h(z_j, \theta_j), \quad j = 0, \dots, N-1, \tag{2}$$
$$z_0 = z,$$

where $z_j$ is the value of the feature variable at the $j$th layer with the total number of layers $N$ and $z = (x_1, \dots, x_n)$. The step size can be chosen in an adaptive manner for each layer and it is not considered as an optimization parameter. The learning problem is to find parameters $\{\theta_j\}_{j=0}^{N-1}$ which lead to the best approximation of $\mathcal{F}$. The parameters are usually found as a solution of the following optimization problem.

$$\min_{\{\theta_j\}} J = L(z_N, y)$$
$$z_{j+1} = \sigma_h(z_j, \theta_j), \quad j = 0, \dots, N-1, \tag{3}$$
$$z_0 = z,$$

where $L(\cdot, \cdot)$ is a loss function which measures the distance between the output of the network and the training data $y = (y_1, \dots, y_N)$.

Because of the connection between network equation 2 and the corresponding neural ODE equation 1, there exists a continuous counterpart of equation 3 which makes equation 3 an approximation of an optimal control problem of the form

$$\min_{\theta(t)} J = L(z(T), y)$$
$$\dot{z}(t) = f(z(t), \theta(t)), \quad t \in [0, T], \tag{4}$$
$$z(0) = z.$$

Solutions of equation 3 are usually found using methods based on gradient descent. Such methods require computations of the gradients of the loss function with respect to all parameters $\{\theta_j\}_{j=0}^{N-1}$.

### 2.2 METHODS OF GRADIENT COMPUTATIONS

**Backpropagation** Let $J = L(z_N, y)$. By the chain rule, the gradient is given by

$$\frac{\partial}{\partial \theta_j} J = \nabla_z L(z_N, y)^\top \frac{\partial z_N}{\partial z_{N-1}} \cdots \frac{\partial z_{j+2}}{\partial z_{j+1}} \frac{\partial z_{j+1}}{\partial \theta_j}.$$

To avoid computationally expensive multiplication of large matrices, the formula is evaluated from the left to the right (backpropagation). This requires that the intermediate values $z_j$ $(j = 1, \dots, N)$ are available. Their size corresponds to the width of the layer and their number to the network's depth $N$.

**Adjoint method** This approach is based on the formula for the continuous gradients via adjoint variables $p(t)$. The adjoint variables are the solution of

$$\dot{p} = -\frac{\partial}{\partial z} f(z(t), \theta(t))^\top p, \qquad p(T) = \nabla L(z(T), y), \tag{5}$$

on time interval $[0, T]$ and the differential of $J(\theta)$ with respect to $\theta(t)$ is calculated as follows

$$DJ(\theta(t)) = p^\top(t) \frac{\partial}{\partial \theta} f(z(t), \theta(t)). \tag{6}$$

This method is particularly interesting when the available memory is limited. In this case, the forward propagation is done to obtain the value of $z(T)$ and $\nabla L(z(T), y)$, there is no need to save the intermediate values $z_j$ and the computational graph, because the backpropagation is realized by integrating the equations for $(z(t), p(t))$ numerically backward in time. The correct discretization of the state-adjoint equations leads to the same expression for the gradient as the one obtained in the backpropagation approach. Still, the values of $z_j$ obtained by the numerical integration backward do not always coincide with the values obtained in the forward pass. This is why this approach usually leads to inexact gradients. This issue can be solved by considering reversible networks.

**Checkpointing** An alternative approach for memory reduction is checkpointing (Gholaminejad et al.), which stores a few intermediate states for a regeneration of the computation graph. In case of adaptive time-stepping it was described in (Zhuang et al., 2020) and further improved in (Matsubara et al., 2021) for the class of Runge-Kutta methods. This is a highly efficient approach, when used in combination with higher order methods. In this case, a small number of checkpoints is required to get a high accuracy in learning. However, when the learning task is to learn a dynamical system from long trajectories of complex systems, then the number of checkpoints becomes large and can lead to memory leaks. This is why it is important to have an alternative approach based on reversible networks with the memory costs independent from the given task.

## 2.3 REVERSIBLE NEURAL NETWORK

A reversible network is a network with the property that there exists an explicit formula for backward propagation $z_{j+1} \mapsto z_j$ that exactly inverts a forward pass $z_j \mapsto z_{j+1}$, i.e., there exists a map $\tilde{\sigma}(\cdot)$, such that $z_j = \tilde{\sigma}(z_{j+1})$ and $z_{j+1} = \sigma_h(\tilde{\sigma}(z_{j+1}), \theta_j)$. It requires that the time-steps $t_0, \ldots, t_N$ have been stored when the forward propagation was computed but it does not require storage of the (potentially very high-dimensional) intermediate values $z_j$. The discretizations of neural ODE equation 2, which admit this property are called reversible methods. As for now, there are only two known reversible methods allowing for an adaptive choice of the step size, namely, asynchronous leapfrog (Zhuang et al., 2021) and reversible Heun (Kidger et al., 2021).

This notion of reversibility for neural networks needs to be contrasted with the notion of time-reversibility or symmetry for numerical integrators. In the context of neural networks, reversibility means that there exists an explicit, efficient formula to invert the forward pass. In numerical integration theory, a time-reversible or symmetric numerical integrator is a formula to advance the solution of an ordinary differential equation by time $h$ such that its inverse is obtained by substituting $h$ by $-h$ (Hairer et al., 2006, II.3). In case of the dynamical system $f(z(t), \theta(t))$ from equation 2, if the discretization by a numerical method $z_{j+1} = \sigma_h(z_j, \theta(t_j))$ is symmetric, then it implies $z_j = \sigma_{-h}(z_{j+1}, \theta(t_{j+1}))$, or equivalently, $z_j = \sigma_{-h}(z_{j+1}, \theta_{j+1})$. Therefore, we can set $\tilde{\sigma}(\cdot) = \sigma_{-h}(\cdot, \theta_{j+1})$, which implies that the method is reversible. The symmetry of integrators is beneficial in the context of the article as inverses of the methods required for backpropagation take simple forms and efficient classical techniques to construct higher order methods (Hairer et al., 2006, II.4) apply.

**Asynchronous Leapfrog (ALF) method** As the optimization parameter $\theta(t)$ in the dynamics $f(z(t), \theta(t))$ depends on time, it can be seen as a part of $f$ and written simply $f_\theta(z(t), t)$. The ALF method requires the augmentation of the pair of state and time $(z, t)$ with the velocity $v$ which approximates $f_\theta(z(t), t)$. We denote a step forward of the ALF method with the step size $h$ by $\Psi_h^{ALF}$. Given a triple $(z_j, v_j, t_j)$ and a step size $h$, the algorithm generates in the forward pass the next values $(z_{j+1}, v_{j+1}, t_{j+1})$ as follows

$$\begin{pmatrix} z_{j+1} \\ v_{j+1} \end{pmatrix} = \Psi_h^{ALF}(z_j, v_j, t_j) = \begin{pmatrix} z_j + h f_\theta(z_j + \frac{h}{2} v_j, t_j + \frac{h}{2}) \\ 2 f_\theta(z_j + \frac{h}{2} v_j, t_j + \frac{h}{2}) - v_j \end{pmatrix}, \quad t_{j+1} = t_j + h. \quad (7)$$

The step backward calculates $(z_j, v_j, t_j)$ from $(z_{j+1}, v_{j+1}, t_{j+1})$ as follows

$$\begin{pmatrix} z_j \\ v_j \end{pmatrix} = \Psi_{-h}^{ALF}(z_{j+1}, v_{j+1}, t_{j+1}), \quad t_j = t_{j+1} - h. \quad (8)$$

If the method is initialized at $(z_0, f(z_0, t_0), t_0)$, then ALF is a second order method in $z$ and first order method in $v$, as it was shown in (Zhuang et al., 2021). The order of accuracy is the order in step size $h$ of the error of the numerical flow compared with the exact flow of the ODE (Hairer et al., 2006). Notice that ALF is symmetric by definition.

The reversible Heun method is another reversible method based on state-space augmentation and was introduced in (Kidger et al., 2021). The method was shown to be also of second order in $z$ and first order in $v$. In addition, it is a symmetric method. In the following part of the paper we will concentrate on the construction of higher order methods based on ALF, but the same can be also applied to the reversible Heun method.

## 3 NEW REVERSIBLE ARCHITECTURES

A general approach in numerical analysis to construct higher order symmetric methods is by composition (Hairer et al., 2006; 2013; Blanes et al., 2024a). In this case one can start with a lower order numerical method and construct a new method by composition of the lower order method with a particular choice of step sizes. This construction leads to a method of higher order of accuracy.

Numerical experiments, see Appendix A, show that ALF is of second order in the error with respect to a high accuracy solver in both $z$ and $v$. This is surprising as the order of consistency of ALF in $v$ is only 1 (Zhuang et al., 2021). Indeed, as we show below, a method consisting of a composition of two steps of ALF has order of consistency 2 in $(z, v)$, which explains the convergence behaviour. This observation is required to apply theory for composition methods (Hairer et al., 2013; Blanes et al., 2024a; Yoshida, 1990) to (two steps of) ALF.

**Theorem 3.1.** *Composition of two steps of ALF methods, i.e. $\Psi_{h/2}^{ALF} \circ \Psi_{h/2}^{ALF}$, applied to $\dot{z} = f(z, t)$ provides second order accurate approximations of position $z$ and velocity $v = \dot{z}$.*

The proof of the theorem is based on comparing the terms in the Taylor series of the exact flow of a differential equation and the numerical flow obtained by composition of two steps of the ALF method. We refer to the Appendix A.1 for the computations. The composition of two steps of the ALF method, each with time-step $\frac{h}{2}$, will be called ALF2 and denoted by $\Psi_h^{ALF2}$. Now we are in a classical situation, with ALF2 a one step reversible method of even order and we can apply the composition methods to construct higher order methods. In this work we consider the Yoshida approach (Yoshida, 1990). Yoshida composition permits to construct methods of a higher accuracy by composing numerical methods of order $2k$ for some integer $k$. It is defined by a symmetric composition of the same method $\Psi^{2\mathrm{k}}$ ($2k$ stands for the order) with different step sizes

$$\Psi_h^Y = \Psi_{ah}^{2\mathrm{k}} \circ \Psi_{bh}^{2\mathrm{k}} \circ \Psi_{ah}^{2\mathrm{k}}$$

with time-steps defined by

$$a = \frac{1}{2 - 2^{\frac{1}{2k+1}}}, \qquad b = 1 - 2a.$$

**Theorem 3.2** ((Yoshida, 1990)). *Yoshida composition of a reversible method $\Psi^{2\mathrm{k}}$ of order $2k$ has order $2k + 2$ and is reversible.*

Yoshida is not the only composition method that can be used, another possible approach is Suzuki composition (Suzuki, 1991). Several approaches are reviewed in (Hairer et al., 2006; Blanes et al., 2024b).

**Remark 3.3.** The approach based on Yoshida composition might require checkpoints in case of certain neural ODEs, e.g., when the learning task is to learn a dispersive partial differential equation such as heat equation. This is because the Yoshida composition forces the use of negative time-steps, which can be a problem in dissipative cases, where it can lead to instability.

In the following, we will denote by $\Psi_h^{Y,2k}$ the higher order methods obtained by Yoshida composition of ALF2, where $2k$ is the order of the method. The constructed higher order reversible method can be used for the construction of a reversible network. In this case, the step forward and the step backward are defined recursively based on the steps forward and backward of a lower order method $\Psi_h^{Y,2k-2}$. The starting method of order 2 is the ALF2 method, i.e. $\Psi_h^{Y,2} = \Psi_h^{ALF2}$ by abuse of notation.

**Adaptive stepping** One of the main advantages in the construction of reversible methods based on ALF is that they allow for adaptive step sizes (Hairer et al., 2006). This can be done in the same manner as for ALF (Zhuang et al., 2021), where the main idea is to delete the computational graph and all the variables needed for the step size computations and only the value of the accepted new step size $h_j$ is saved. As a result, values $h_1, \ldots, h_N$ are saved and then accessed in the integration

---

**Algorithm 1** Step forward of $2k$-th order Yoshida

---

1. Input: $(z_j, v_j, t_j, h_j)$

2. Set $a = 1/(2 - 2^{\frac{1}{2k+1}})$, $b = 1 - 2a$.

3. Set $(\tilde{z}_1, \tilde{v}_1) = \Psi^{Y,2k-2}_{ah_j}(z_j, v_j, t_j)$, $\tilde{t}_1 = t_j + ah_j$

4. Set $(\tilde{z}_2, \tilde{v}_2) = \Psi^{Y,2k-2}_{bh_j}(\tilde{z}_1, \tilde{v}_1, \tilde{t}_1)$, $\tilde{t}_2 = \tilde{t}_1 + bh_j$

5. Set $(z_{j+1}, v_{j+1}) = \Psi^{Y,2k-2}_{ah_j}(\tilde{z}_2, \tilde{v}_2, \tilde{t}_2)$, $t_{j+1} = \tilde{t}_2 + ah_j$

**if** adaptive time-stepping **then**

    6a. compute the error of $z_{j+1}, v_{j+1}$ w.r.t. the output of a $(2k+1)$st order integration method

    6b. compute the new $h_{j+1}$ following (Hairer et al., 2006)

**else**

    6. $h_{j+1} = h_j$

**end if**

7. Output: $(z_{j+1}, v_{j+1}, t_{j+1}, h_{j+1})$

---

backward needed for gradient computations. This can be done in exactly the same manner for our Yoshida-based methods. The resulting steps forward and backward are summarized in Algorithm 1 and Algorithm 2.

**Remark 3.4.** Notice that even though the reversible Heun method was proved to be of order $(2, 1)$ in $(z, v)$ in (Kidger et al., 2021), it was noted in the same paper that it gains the second order in both variables at even steps. This implies that the composition approach can be used in this case as well.

---

**Algorithm 2** Step backward of $2k$-th order Yoshida

---

1. Input: $(z_{j+1}, v_{j+1}, t_{j+1}, h_{j+1})$

2. Set $a = 1/(2 - 2^{\frac{1}{2k+1}})$, $b = 1 - 2a$.

3. Set $(\tilde{z}_1, \tilde{v}_1) = \Psi^{Y,2k-2}_{-ah_{j+1}}(z_{j+1}, v_{j+1}, t_{j+1})$, $\tilde{t}_1 = t_{j+1} - ah_{j+1}$

4. Set $(\tilde{z}_2, \tilde{v}_2) = \Psi^{Y,2k-2}_{-bh_{j+1}}(\tilde{z}_1, \tilde{v}_1, \tilde{t}_1)$, $\tilde{t}_2 = \tilde{t}_1 - bh_{j+1}$

5. Set $(z_j, v_j) = \Psi^{Y,2k-2}_{-ah_{j+1}}(\tilde{z}_2, \tilde{v}_2, \tilde{t}_2)$, $t_j = \tilde{t}_2 - ah_{j+1}$

6. Set $h_j$ from $h_1, \ldots, h_N$ obtained in the integration forward

7. Output: $(z_j, v_j, t_j, h_j)$

---

**Gradient computations**  The augmentation of the feature space leads to the new variable which we denote by $\phi = (z, v)$. Then, the learning problem is formulated as follows with $P_z(\phi)$ projection of $\phi$ to $z$

$$\min_{\{\theta_j\}} J = L(P_z(\phi_N), y)$$

$$\phi_{j+1} = \Psi^{Y,2k}_h(\phi_j, \theta_j), \quad j = 0, \ldots, N - 1, \tag{9}$$

$$\phi_0 = (z, f(z, \theta_0)).$$

Following (Griesse & Walther, 2004), the discrete version of equation 5 associated with equation 9 is given by

$$(\lambda_N)^\top = \nabla L(\phi_N), \quad \lambda_j = \left(\frac{\partial \phi_{j+1}}{\partial \phi_j}\right)^\top \lambda_{j+1}, \tag{10}$$

and the gradients are computed by

$$\frac{\partial J(\theta)}{\partial \theta_j} = \lambda_{j+1}^\top \frac{\partial \phi_{j+1}}{\partial \theta_j}. \tag{11}$$

The adjoint method for the gradient computation as in the MALI network (Zhuang et al., 2021) and the reversible Heun network (Kidger et al., 2021) is based on the propagation $\phi_j \to \phi_{j+1}$ and automatic

differentiation for the computation of the step backward of the adjoint variable following equation 10. The resulting method of gradient computation is summarized in Algorithm 3. Alternatively, the exact expression of the numerical method governing the adjoint dynamics equation 10 can be obtained, see details in Appendix B. In this case, there is no need to compute $\frac{\partial \phi_{j+1}}{\partial \phi_j}$, which makes the approach computationally more efficient and memory efficient.

---

**Algorithm 3** Computation of gradients

    1. Input: training data $z_0$, initialization of parameters $\theta$, velocity $v_0 = f(z_0, \theta_0)$
    2. Propagate through the network using $\Psi_h^{Y,2k}$ to get $(z_N, v_N)$
    3. Set $\lambda_N^z = \nabla L(z_N, y)$ and $\lambda_N^v = 0$
    **for** j = N-1 to 1 **do**
        4. Compute $\phi_j$ from $\phi_{j+1}$ using Algorithm 2
        5. Compute $\phi_{j+1}$ from $\phi_j$ using Algorithm 1 to get the computational graph
        6. Compute $\lambda_j$ from $\lambda_{j+1}$ using equation 10 and AD to compute $\frac{\partial \phi_{j+1}}{\partial \phi_j}$
        7. Compute $\frac{\partial J(\theta)}{\partial \theta_j}$ using equation 11
        8. Delete $\lambda_{j+1}, \phi_{j+1}$ and the computational graphs
    **end for**
    9. Output: gradients $\frac{\partial J(\theta)}{\partial \theta_j}$ for $j = 1, \ldots, N-1$.

---

**Costs comparison** We will use the following notations: $d$ is the dimension of $z$, $T$ is the length of the time interval in the continuous-depth setting, $N$ is the number of layers, $M$ stands for the number of layers in $f$, when $f$ is given by a neural network itself, $s$ denotes the number of steps needed for the computation of a time-step in the adaptive step size selection, $p$ is the order of the considered numerical method and $r$ is the number of evaluations of $f$ used in the numerical method (e.g. stages in Runge-Kutta methods or compositions in our approach). We show the comparison of the new proposed approach with the standard backpropagation approach, adjoint method version NODE (Chen et al., 2018), ACA approach (Zhuang et al., 2020) and MALI approach (Zhuang et al., 2021) in Table 1, which extends the Table 1 in (Zhuang et al., 2021). We use big $\mathcal{O}$ notation, when the constants depend on the learning tasks.

**Computational costs** The compositional structure of the proposed method directly implies that the computation costs for gradient computations are equal to the computational costs by ALF multiplied by $r$, the number of the compositions. Notice that $N$ depends on the order $p$ of the discretization method and becomes smaller when the order is higher for fixed $\varepsilon$ and $T$. As a result, ALF method needs more time-steps, than higher order methods for $\varepsilon < 1$, which is related to the bias in the learned parameters in the task of identification of the parameters, as explained in Appendix C.1 and illustrated in Figure 4, and to the training error.

**Memory costs** The gradient computation requires to compute $\frac{\partial \phi_{j+1}}{\partial \phi_j}$ leading to the storage of all the intermediate states involved in the step forward. This increases the memory costs of MALI by a factor $r$, see Table 1. Notice that the approach presented in Appendix B does not require to store all the intermediate states. Indeed, a step backward of the state-adjoint system is a composition of rescaled steps backward of the ALF method. Therefore, we only need to store one intermediate state obtained in the composition at a time. This makes the method of the same memory cost as MALI. Notice that depending on the depth of the network, adaptive checkpointing as in (Matsubara et al., 2021) can be added. When no checkpoints are needed the behaviour is as in NODE and in the worst case the behaviour is as in the backprop. In general, the number of checkpoints depends on $N$, which depends linearly on $T$. Therefore, more checkpoints are needed in case of large $T$.

## 4 EXPERIMENTS

### 4.1 PARAMETER IDENTIFICATION IN DYNAMICAL SYSTEMS

We consider the identification problem of unknown parameters of a dynamical system. The structure of the differential equations is assumed to be known, but some parameters in the equations are unknown. The training data is given by snapshots of trajectories $\{x_l(t_i)\}_{i,l}$ with $l = 1, \ldots L$, $i = 0, \ldots, I$. The goal is to learn the parameters from the given trajectories. This class of problems can

Table 1: Comparison of costs in gradient computations for different approaches

| Method | Computational costs | Memory costs | Number of epochs $N$ in function of accuracy $\varepsilon^*$ |
|---|---|---|---|
| Backprop | $r \times d \times M \times N \times s \times 2$ | $r \times d \times M \times N \times s$ | $T \times \mathcal{O}(\varepsilon^{-\frac{1}{p+1}})$ |
| NODE | $r \times d \times M \times N \times s \times 2$ | $d \times M$ | $T \times \mathcal{O}(\varepsilon^{-\frac{1}{p+1}})$ |
| ACA | $r \times d \times M \times N \times (s+1)$ | $d \times (M+N)$ | $T \times \mathcal{O}(\varepsilon^{-\frac{1}{p+1}})$ |
| MALI | $d \times M \times N \times (s+2)$ | $d \times (M+1)$ | $T \times \mathcal{O}(\varepsilon^{-\frac{1}{3}})$ |
| Proposed method | $r \times d \times M \times N \times (s+2)$ | $r^{**} \times d \times (M+1)$ | $T \times \mathcal{O}(\varepsilon^{-\frac{1}{p+1}})$ |

$^*$ $\varepsilon$ is the error tolerance for the estimation of the local error in the stepsize selection
$^{**}$ Memory costs of the proposed method can be reduced to $r = 1$, if the gradients are computed as in
AppendixB

be naturally treated using the neural ODE approach. The vector field $f$ in equation 1 is given by the known differential equation and $\theta = \theta_1, \ldots, \theta_s$ is the set of unknown parameters. In this case, the learning problem can be stated in the form of equation 9, where the same $\theta_1, \ldots, \theta_s$ appear all at each layer. The training data $z_0 = (x_1(t_0), \ldots, x_L(t_0))$ stands for the initial points and $y$ includes all the other points of the given trajectories. We denote by $y(t_i)$ the points in $y$ corresponding to trajectories at time $t_i$ for $i = 1, \ldots, I$. The loss have a particular structure in this case as it depends on the intermediate states obtained during the integration of neural ODE, namely, it depends on $(z_{N_1}, \ldots, z_{N_I})$, to measure the distance with the given trajectories points $(y(t_1), \ldots, y(t_I))$. As a result, it takes the form $L = \sum_{i=1}^{I} L_i(z_{N_i}, y(t_i))$. Because of the additive form of the loss, the gradients can be computed as a sum of the corresponding gradients of $L_1, \ldots, L_I$ as follows

$$\frac{\partial L}{\partial \theta_i} = \frac{\partial L_1}{\partial \theta_i} + \cdots + \frac{\partial L_I}{\partial \theta_i}, \qquad i = 1, \ldots, s,$$

where each of the terms in the sum is computed using Algorithm 3. The memory efficiency is still important in this case, because we do not store all the intermediate states at the propagation forward, but only the states which approximate the trajectories at the desired times $t_1, \ldots, t_I$.

**Statistical inference**  In simulation based inference or likelihood-free inference probabilistic methods are employed to identify parameters in models based on repeated forward simulations (Cranmer et al., 2020a). Traditionally, these consider the forward pass as a black box (such as Approximate Bayesian Computation (ABC) (Rubin, 1984; Beaumont et al., 2002)) and do not require differentiability with respect to the model parameters or the inputs. This needs to be contrasted to our proposed neural network architecture, which is designed to circumvent large memory requirements in the computation of gradients when the layers are wide. Indeed, a combination of our architecture with simulation based inference models that do make use of gradients such as (Graham & Storkey, 2017) constitutes an interesting avenue for future research.

### 4.1.1 KEPLER PROBLEM

We consider the Kepler problem, where the dynamics describes the evolution of the position $q$ and velocity $v$ of a mass point moving around a much heavier body. It is modeled on the 4-dimensional space $x = (q, v) \in \mathbb{R}^2 \times \mathbb{R}^2$. The equations are defined on the time interval $[0, 1]$ as follows

$$\dot{q} = v, \quad \dot{v} = -\frac{\alpha}{\|q\|^3} q, \tag{12}$$

with an unknown parameter $\alpha \in \mathbb{R}$. The training set is given by the initial condition $x(t_0)$ and $q$-coordinate of 5 points on a trajectory of equation 12 generated with $\alpha = \pi/4 \approx 0.785$, i.e. $\{q(t_i)\}_{i=1}^5$. The task is to learn $\alpha$ as accurately as possible. From the training set we form $z_0 = x(t_0)$ and the corresponding $y(t_i) = q(t_i)$ for $i = 1, \ldots, 5$. We set up a learning problem in the form of equation 9 with the loss defined by $L = \sum_{i=1}^{5} \|q_{N_i} - q(t_i)\|^2$ with $q_{N_i}$ projection of $z_{N_i}$ to $q$-coordinate and $N_i$ the number of time-steps used in the integration from $t_{i-1}$ to $t_i$. We compare two algorithms for performing the numerical integration during training, namely, ALF and the Yoshida composition of ALF2 of order 4. We write Y4 for the Yoshida composition method for shortness. We test the wall-clock time required to reach loss accuracy $10^{-8}$ using adaptive methods. The tests are run for different initializations of $\alpha$ in optimization. The results can be seen in Table 2. It can be observed that in all tests, Y4 is at least four times faster than ALF. The reason of the faster training for Y4 is

Table 2: Time to reach accuracy $10^{-8}$ using adaptive methods in the Kepler problem

| Initial value of $\alpha$ | Computation time | |
| --- | --- | --- |
| | adaptive ALF | adaptive Y4 |
| 0.1 | 7.68 sec | 2.42 sec |
| 0.7 | 4.07 sec | 1.02 sec |
| 0.75 | 3.26 sec | 0.803 sec |
| 0.8 | 2.5 sec | 0.44 sec |
| 1.3 | 8.39 sec | 3.85 sec |

Table 3: Time to reach accuracy $10^{-4}$ using adaptive methods in nonlinear oscillators problem

| Mean parameter error at initialization | Computation time | |
| --- | --- | --- |
| | adaptive ALF | adaptive Y4 |
| 0.28897009 | 81608 sec | 51720 sec |
| 0.29821727 | 68645 sec | 40646 sec |
| 0.30549358 | 56764 sec | 29990 sec |
| 0.30289593 | 96301 sec | 46524 sec |
| 0.29106813 | 22161 sec | 13790 sec |

in using larger step sizes for the forward integration. The lower order method requires smaller step sizes to reach the same accuracy defined by an error tolerance and this leads to more steps in the computation of trajectories. Additional results for the Kepler problem supporting the reasoning can be found in Appendix C.1.

### 4.1.2 NONLINEAR HARMONIC OSCILLATOR

In the second example we consider a system of coupled Duffing oscillators, which describes the movements of a coupled system of mass points attached with springs with nonlinear elastic forces. The dynamics of $N$ mass points is given by the following equations

$$\dot{q}_i = v_i, \quad \dot{v}_i = -a_i q_i - b_i q_i^3 - \sum_{j=1}^{N} e_{i,j}(q_i - q_j), \qquad i = 1, \ldots, N, \tag{13}$$

with the condition $e_{i,j} = e_{j,i}$. Positions of $N$ mass points are given by $q = (q_1, \cdots, q_N) \in \mathbb{R}^N$ and velocities by $v = (v_1, \cdots, v_N) \in \mathbb{R}^N$. We set $x = (q, v) \in \mathbb{R}^{2N}$. In the numerical experiments we fix $N = 10$ and assume that parameters $a_i, b_i, e_{i,j} \in \mathbb{R}$ for $i, j = 1, \ldots, 10$ are unknown. As a result, equation 13 has dimension 20 with 65 unknown parameters. The training set consists of initial and final positions of 200 trajectories, that is $z_0 = (x_1(t_0), \ldots, x_{200}(t_0))$ and $y = y(t_1) = (x_1(t_1), \ldots, x_{200}(t_1))$. In this setting, we compare the computational time to reach a certain training accuracy of ALF and Y4 with adaptive time-stepping and the training accuracy of ALF and Y4 with fixed step-size. We present the wall-clock times to reach the training accuracy $10^{-4}$ in Table 3. The time required by Y4 to reach accuracy $10^{-4}$ is almost two times smaller which illustrates the lower computational costs of the method. As before, the ALF method with adaptive time-stepping requires smaller step sizes and more steps are used in each epoch of the optimization. Details with additional results confirming the behaviour are presented in Appendix C.2.1.

### 4.2 LEARNING OF DYNAMICAL SYSTEMS PARAMETERIZED BY NEURAL NETWORK

We consider a problem, where a part of the structure of the differential equations is known and the unknown part is approximated using a neural network. Our goal is to find the neural network parameterization such that the resulting trajectories of the system are as close as possible to given trajectories from the training set. As before, the problem can be treated by the neural ODE approach. In this case the vector field $f$ in equation 1 is given by a neural network.

### 4.2.1 NONLINEAR HARMONIC OSCILLATOR

We consider the problem of approximating the potential function of a physical system, given by the Duffing oscillators with two mass points. Equations can be equivalently written as

$$\dot{q} = v, \quad \dot{v} = -\nabla V(q), \tag{14}$$

with $q = (q_1, q_2)$, $v = (v_1, v_2)$ and $V(q)$ stands for the potential energy of the system. The learning task is to learn $V(q)$. The gradient of the potential is approximated using a neural network with 51500

Table 4: Time to get accuracy $10^{-2}$ by adaptive methods in oscillators problem parameterized by NN

| Computation time | | |
|---|---|---|
| Random initialization of parameters in NN | adaptive ALF | adaptive Y4 |
| Initialization 1 | 2504 sec | 1974 sec |
| Initialization 2 | 2961 sec | 1857 sec |
| Initialization 3 | 4185 sec | 2627 sec |
| Initialization 4 | 3542 sec | 2125 sec |
| Initialization 5 | 3396 sec | 2616 sec |

Table 5: Wallclock time to get the training loss below $10^{-3}$ by adaptive methods in discretized PDE

| Computation time | | |
|---|---|---|
| Random initialization of parameters in NN | adaptive ALF | adaptive Y4 |
| Initialization 1 | 336.6808 sec | 138.5936 sec |
| Initialization 2 | 169.5010 sec | 133.6825 sec |
| Initialization 3 | 180.8185 sec | 140.3172 sec |
| Initialization 4 | 153.7389 sec | 128.2795 sec |
| Initialization 5 | 142.7732 sec | 142.9196 sec |

parameters. To obtain the potential from the learned vector field, we apply numerical integration methods to the neural network approximating $-\nabla V(q)$. We compare the computational time of ALF and Y4 to reduce the value of the loss function below $10^{-2}$. The results presented in Table 4 show that Y4 is faster than ALF in completing the training on different random initializations of the network parameters.

### 4.2.2 DISCRETIZED WAVE EQUATION

In the second example we consider the 1-dimensional wave equation $u_{tt}(t, x) = u_{xx}(t, x) - \nabla V(u(t, x))$ on the spatial-temporal domain $[0, 1] \times [0, 0.3]$ with periodic boundary conditions in space for the potential $V(u) = \frac{1}{2}u^2$. On a spatial, equidistant, periodic mesh with mesh width $\Delta x = \frac{1}{40}$ we seek to describe the system's evolution by the first order system

$$\dot{u}_d = v_d, \quad \dot{v}_d = f(u_d), \tag{15}$$

where the unknown function $f$ is parametrized as a fully connected ReLU neural network with one hidden layer of size 100. The dimension of $(u_d, v_d)$ is 40. We compare the training performance of ALF and Y4. Both adaptive methods are employed with the same error tolerance. Yoshida is faster in finishing each epoch and the optimizer takes less time to minimize the training loss below $10^{-3}$. The precise results are reported in Table 5 for 5 random initializations in the training. This illustrates the applicability of our method to the highly active research area of learning models of systems that are governed by partial differential equations.

## 5 CONCLUSION

In this work, we construct higher order reversible methods. These constitute explicit numerical integrators which are compatible with adaptive step-size selection strategies. The methods are employed to train deep neural networks that are based on neural ODEs. Thanks to the reversibility property, we avoid high memory requirements for backpropagation in the optimization procedure of the network parameters. Memory efficient backpropagation allows an application of deep architectures to the identification tasks of models of high-dimensional dynamical systems, which arise, for instance, as spatial discretizations of partial differential equations. As the method is based on neural ODEs, it can be trained with time-series data at irregular time-steps and can predict continuous time-series data. We showed the advantages of the newly constructed networks on the example of a network based on a 4th order method and demonstrate lower memory costs and faster training in comparison to lower order methods.

While the examples in the article focus on system identification tasks for systems governed by differential equations, extensions to neural stochastic differential equations (Kidger et al., 2021) are of interest and applications to normalizing flows or image processing (Allen-Blanchette et al., 2020) can be an exciting avenue to explore in future works.

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

## A  ERROR ANALYSIS OF ALF2

**Example A.1.** Consider a simple example of a differential equation on $\mathbb{R}$ given by

$$\dot{z} = z^2 + t + \sin(zt) + \frac{1}{z^2 + 1}. \tag{16}$$

We solve the equation numerically using the ALF method and compare with a solution of high accuracy for different step sizes. The results are plotted in Figure 1 and show the second order behaviour in both $(z, v)$ variables.

### A.1  PROOF OF THEOREM 3.1

We show that the local error of ALF2 in $(z, v)$ is of order $O(h^3)$. Let us consider the Taylor expansion of the exact flow $(z(t), v(t))$ around $(z(t_0), v(t_0) = f(z_0, t_0))$.

$$z(t_0 + h) = z_0 + hf(z_0, t_0) + \frac{h^2}{2}\left(\frac{\partial f}{\partial z}(z_0, t_0) \circ f(z_0, t_0) + \frac{\partial f}{\partial t}(z_0, t_0)\right) + O(h^3),$$

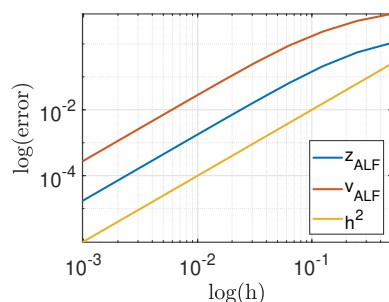

Figure 1: Log-log plot of the global error of trajectories $(z(t), v(t))$ of equation 16 defined on time interval $[0, 1.0]$ and obtained by ALF with $h$ ranging from $0.5$ to $10^{-3}$.

$$v(t_0 + h) = f(z_0, t_0) + h\left(\frac{\partial f}{\partial z}(z_0, t_0) \circ f(z_0, t_0) + \frac{\partial f}{\partial t}(z_0, t_0)\right) +$$

$$\frac{h^2}{2}(\frac{\partial^2 f}{\partial z^2}(z_0, t_0)(f(z_0, t_0), f(z_0, t_0)) + \frac{\partial f}{\partial z}(z_0, t_0) \circ \frac{\partial f}{\partial z}(z_0, t_0) \circ f(z_0, t_0) + 2\frac{\partial^2 f}{\partial z \partial t}(z_0, t_0) \circ f(z_0, t_0)$$

$$+ \frac{\partial f}{\partial z}(z_0, t_0) \circ \frac{\partial f}{\partial t}(z_0, t_0) + \frac{\partial^2 f}{\partial t^2}(z_0, t_0)) + O(h^3).$$

Now we consider the same for the numerical flow obtained with ALF2, that is composition of two steps of ALF each with the step size $\frac{h}{2}$. One step of ALF2 from $(z_0, v_0)$ leads to $(z_1, v_1)$ of the form

$$z_1(h) = z_0 + \frac{h}{2}(f(z_0 + \frac{h}{4}f(z_0, t_0), t_0 + \frac{h}{4}) + f(z_0 + hf(z_0 + \frac{h}{4}f(z_0, t_0), t_0 + \frac{h}{4})$$

$$- \frac{h}{4}f(z_0, t_0), t_0 + \frac{3h}{4})),$$

and

$$v_1(h) = v_0 + 2(f(z_0 + hf(z_0 + \frac{h}{4}f(z_0, t_0), t_0 + \frac{h}{4}) - \frac{h}{4}v_0, t_0 + \frac{3h}{4}) - f(z_0 + \frac{h}{4}f(z_0, t_0), t_0 + \frac{h}{4})).$$

Writing down the Taylor expansion in $h$ for $(z_1(h), v_1(h))$ we find exactly the same terms as in $(z(t_0 + h), v(t_0 + h))$ up to terms of the third order $O(h^3)$. The computations to obtain the Taylor expansion of $(z_1, v_1)$ were done using Maple software. This implies that the local error of ALF2 is of the 3rd order, and therefore, the global error is of order 2. This completes the proof.

## B   NUMERICAL METHOD FOR THE ADJOINT

The expression of equation 10 for $\phi_{k+1}$ obtained from $\phi_k$ by a step forward of ALF2 can be interpreted as a rescaled step backward of ALF2 applied to the state-adjoint dynamics. Let us introduce a map $W_h$ depending on $h$, which acts on $(z, v, \lambda^z, \lambda^v)$ as follows. It only transforms $\lambda^v$ multiplying it by $-\frac{h^2}{16}$, that is

$$W_\alpha(z, v, \lambda^z, \lambda^v) = \begin{pmatrix} \text{Id} & 0 & 0 & 0 \\ 0 & \text{Id} & 0 & 0 \\ 0 & 0 & \text{Id} & 0 \\ 0 & 0 & 0 & \alpha\text{Id} \end{pmatrix} \begin{pmatrix} z \\ v \\ \lambda^z \\ \lambda^v \end{pmatrix}.$$

**Theorem B.1.** *The step backward of the discretized state-adjoint system associated to the ALF2 method satisfies*

$$(z_k, v_k, \lambda_k^z, \lambda_k^v) = W_{\frac{-h^2}{16}}^{-1} \circ \Psi_{-h}^{ALF2} \circ W_{\frac{-h^2}{16}}(z_{k+1}, v_{k+1}, \lambda_{k+1}^z, \lambda_{k+1}^v), \tag{17}$$

*where $\Psi_{-h}^{ALF2}$ is applied to the state-adjoint equations of the augmented system for $\phi = (z, v)$*

$$\dot{\phi}(t) = \tilde{f}(\phi(t), \theta(t)), \quad \dot{\lambda} = -\frac{\partial}{\partial \phi}\tilde{f}(\phi(t), \theta(t))^\top \lambda,$$

*with $\tilde{f}(\phi, \theta) = (f(z, \theta), \frac{\partial}{\partial z}f(z, \theta)f(z, \theta))$.*

*Proof.* In order to find the expression for ALF2, we first determine the expression for ALF and use the chain rule. Let us compute $\frac{\partial \phi_{k+1}}{\partial \phi_k}$ for ALF method, where $\phi_{k+1} = (z_{k+1}, v_{k+1})$ and $\phi_k = (z_k, v_k)$. Differentiating equation 7 with respect to $(z_k, v_k)$, we obtain

$$
\frac{\partial \phi_{k+1}}{\partial \phi_k} = \begin{pmatrix} \mathrm{Id} + h\frac{\partial f}{\partial z}(z_k + \frac{h}{2}v_k, t_k + \frac{h}{2}) & \frac{h^2}{2}\frac{\partial f}{\partial z}(z_k + \frac{h}{2}v_k, t_k + \frac{h}{2}) \\ 2\frac{\partial f}{\partial z}(z_k + \frac{h}{2}v_k, t_k + \frac{h}{2}) & h\frac{\partial f}{\partial z}(z_k + \frac{h}{2}v_k, t_k + \frac{h}{2}) - \mathrm{Id} \end{pmatrix}.
$$

This implies

$$
\begin{aligned}
\lambda_k^z &= \left( \mathrm{Id} + h\frac{\partial}{\partial z}f(z^k + \frac{h}{2}v^k, t^k + \frac{h}{2}) \right) \lambda_{k+1}^z + 2\frac{\partial}{\partial z}f(z^k + \frac{h}{2}v^k, t^k + \frac{h}{2})\lambda_{k+1}^v, \\
\lambda_k^v &= \frac{h^2}{2}\frac{\partial}{\partial z}f(z^k + \frac{h}{2}v^k, t^k + \frac{h}{2})\lambda_{k+1}^z + \left( h\frac{\partial}{\partial z}f(z^k + \frac{h}{2}v^k, t^k + \frac{h}{2}) - \mathrm{Id} \right)\lambda_{k+1}^v.
\end{aligned}
\tag{18}
$$

Notice that equation 18 can be equivalently written as

$$
\begin{aligned}
\lambda_k^z &= \lambda_{k+1}^z + h\frac{\partial}{\partial z}f(z^k + \frac{h}{2}v^k, t^k + \frac{h}{2})\left( \lambda_{k+1}^z + \frac{2}{h}\lambda_{k+1}^v \right), \\
\lambda_k^v &= -2\frac{\partial}{\partial z}f(z^k + \frac{h}{2}v^k, t^k + \frac{h}{2})\left( -\frac{h^2}{4}\lambda_{k+1}^z - \frac{h}{2}\lambda_{k+1}^v \right) - \lambda_{k+1}^v.
\end{aligned}
\tag{19}
$$

Let us now introduce $\tilde{\lambda}_k^v = -\frac{4}{h^2}\lambda_k^v$. Then equations take the following form

$$
\begin{aligned}
\lambda_k^z &= \lambda_{k+1}^z + h\frac{\partial}{\partial z}f(z^k + \frac{h}{2}v^k, t^k + \frac{h}{2})\left( \lambda_{k+1}^z - \frac{h}{2}\tilde{\lambda}_{k+1}^v \right), \\
\tilde{\lambda}_k^v &= -2\frac{\partial}{\partial z}f(z^k + \frac{h}{2}v^k, t^k + \frac{h}{2})\left( \lambda_{k+1}^z - \frac{h}{2}\tilde{\lambda}_{k+1}^v \right) - \tilde{\lambda}_{k+1}^v.
\end{aligned}
\tag{20}
$$

Taking into account that $z_k + \frac{h}{2}v_k = z_{k+1} - \frac{h}{2}v_{k+1}$ from the construction of equation 7-equation 8, we conclude that variables $(\lambda_k^z, -\frac{4}{h^2}\lambda_k^v)$ follow the backward integration with ALF method and its step backward defined by equation 8 applied to the continuous equations of the adjoint equation 5. As a result, the step backward of the adjoint variables $\lambda$ can be expressed as

$$
(\lambda_k^z, \lambda_k^v) = \widehat{W}_{\frac{-h^2}{4}}^{-1} \circ \widehat{\Psi}_{-h}^{ALF}(z_{k+1}, v_{k+1}) \circ \widehat{W}_{\frac{-h^2}{4}}(\lambda_{k+1}^z, \lambda_{k+1}^v),
$$

where $\widehat{W}_\alpha$ is a projection of $W_\alpha$ to variables $(\lambda_k^z, \lambda_k^v)$ and $\widehat{\Psi}_{-h}^{ALF}(z_{k+1}, v_{k+1})$ stands for a projection of the backward ALF step to $(\lambda^z, \lambda^v)$, which is still a function of $(z_{k+1}, v_{k+1})$. To deduce the formula for the ALF2 method, we use its composition structure, namely, $\Psi_h^{ALF2} = \Psi_{h/2}^{ALF} \circ \Psi_{h/2}^{ALF}$. This implies

$$
\frac{\partial \phi_{k+1}}{\partial \phi_k} = \frac{\partial}{\partial \phi_k}\left( \Psi_{h/2}^{ALF} \circ \Psi_{h/2}^{ALF} \right) = \left( \frac{\partial \Psi_{h/2}^{ALF2}}{\partial \phi}(\phi_{k+\frac{1}{2}}) \right) \circ \left( \frac{\partial \Psi_{h/2}^{ALF2}}{\partial \phi}(\phi_k) \right)
$$

with

$$
\phi_{k+\frac{1}{2}} = \Psi_{h/2}^{ALF2}(\phi_k) = \Psi_{-h/2}^{ALF2}(\phi_{k+1}).
$$

As a result,

$$
\begin{aligned}
\left( \frac{\partial \phi_{k+1}}{\partial \phi_k} \right)^\top &= \left( \frac{\partial \Psi_{h/2}^{ALF2}}{\partial \phi}(\phi_k) \right)^\top \circ \left( \frac{\partial \Psi_{h/2}^{ALF2}}{\partial \phi}(\phi_{k+\frac{1}{2}}) \right)^\top \\
&= \widehat{W}_{\frac{-h^2}{16}}^{-1} \circ \widehat{\Psi}_{-h/2}^{ALF}(\phi_{k+\frac{1}{2}}) \circ \widehat{W}_{\frac{-h^2}{16}} \circ \widehat{W}_{\frac{-h^2}{16}}^{-1} \circ \widehat{\Psi}_{-h/2}^{ALF}(\phi_{k+1}) \circ \widehat{W}_{\frac{-h^2}{16}} \\
&= \widehat{W}_{\frac{-h^2}{16}}^{-1} \circ \widehat{\Psi}_{-h/2}^{ALF}(\Psi_{-h/2}^{ALF2}(\phi_{k+1})) \circ \widehat{\Psi}_{-h/2}^{ALF}(\phi_{k+1}) \circ \widehat{W}_{\frac{-h^2}{16}} \\
&= \widehat{W}_{\frac{-h^2}{16}}^{-1} \circ \widehat{\Psi}_{-h}^{ALF2}(z_{k+1}, v_{k+1}) \circ \widehat{W}_{\frac{-h^2}{16}}.
\end{aligned}
$$

The resulting equations for the backward step of the state-adjoint system are given in equation 17. This completes the proof of Theorem B.1. □

The formula for the adjoint of Yoshida methods $\Phi_{2k}^Y$ follows from the composition structure of the method and is presented the following theorem. With a slight abuse of notation we denote $\Phi^{ALF2}$ by $\Phi_2^Y$.

**Theorem B.2.** *Assume that the discrete one step method in equation 2 is given for $k \geq 2$ by*

$$\Psi_{2k}^Y(h) = \Psi_{2k-2}^Y(ah) \circ \Psi_{2k-2}^Y(bh) \circ \Psi_{2k-2}^Y(ah), \qquad \phi_{k+1} = \Psi_{2k}^Y(h) \circ \phi_k. \tag{21}$$

*Then the state-adjoint backward step can be computed recursively as follows*

$$(\phi_k, \lambda_k) = \widetilde{\Psi}_{2k-2}^Y(ah) \circ \widetilde{\Psi}_{2k-2}^Y(bh) \circ \widetilde{\Psi}_{2k-2}^Y(ah)(\phi_{k+1}, \lambda_{k+1}), \tag{22}$$

*with $\widetilde{\Psi}_{2k-2}^Y$ the map, which defines the backward step of state-adjoint system of the method $\Phi_{2k-2}^Y$.*

*Proof.* The proof is by induction on $k$ in the considered method $\Psi_{2k}^Y$ and is based on the composition structure of $\Psi_{2k}^Y$ in equation 21. Let $k = 2$, then $\Psi_4^Y(h) = \Psi_{ah}^{ALF2} \circ \Psi_{bh}^{ALF2} \circ \Psi_{ah}^{ALF2}$. By the chain rule we have

$$\left(\frac{\partial \phi_{k+1}}{\partial \phi_k}\right)^\top = \left(\frac{\partial \Psi_{ah}^{ALF2}}{\partial \phi}(\phi_{k+\frac{1}{3}})\right)^\top \circ \left(\frac{\partial \Psi_{bh}^{ALF2}}{\partial \phi}(\phi_{k+\frac{2}{3}})\right)^\top \circ \left(\frac{\partial \Psi_{ah}^{ALF2}}{\partial \phi}(\phi_{k+1})\right)^\top$$

with

$$\phi_{k+\frac{2}{3}} = \Psi_{-ah}^{ALF2}(\phi_{k+1}),$$
$$\phi_{k+\frac{1}{3}} = \Psi_{-bh}^{ALF2} \circ \Psi_{-ah}^{ALF2}(\phi_{k+1}).$$

By construction of the backward step of the state adjoint system by ALF2 shown in equation 17, we have

$$\left(\frac{\partial \phi_{k+1}}{\partial \phi_k}\right)^\top = Pr_\lambda \left(W_{ah}^{-1} \circ \Psi_{-ah}^{ALF2} \circ W_{ah}\right) \circ Pr_\lambda \left(W_{bh}^{-1} \circ \Psi_{-bh}^{ALF2} \circ W_{bh}\right) \circ$$
$$\circ Pr_\lambda \left(W_{ah}^{-1} \circ \Psi_{-ah}^{ALF2} \circ W_{ah}\right),$$

where $\widehat{\Psi}_{h_i}^{ALF2} = Pr_\lambda \left(W_{h_i}^{-1} \circ \Psi_{-h_i}^{ALF2} \circ W_{h_i}\right)$, $h_i \in \{ah, bh\}$ defines a step backward with the ALF2 method with step-size $h_i$ in the adjoint variable. This proves the Theorem for $k = 2$. Let us assume now that the statement of the theorem holds for $k = k_0$ and we consider the adjoint method for $\Psi_{2k_0}^Y(h) = \Psi_{2k_0-2}^Y(ah) \circ \Psi_{2k_0-2}^Y(bh) \circ \Psi_{2k_0-2}^Y(ah)$. As before, applying the chain rule and the assumption of the induction, it follows that

$$\left(\frac{\partial \phi_{k+1}}{\partial \phi_k}\right)^\top = \left(\frac{\partial \Psi_{2k_0-2}^Y(ah)}{\partial \phi}(\tilde{\phi}_{k+\frac{1}{3}})\right)^\top \circ \left(\frac{\partial \Psi_{2k_0-2}^Y(bh)}{\partial \phi}(\tilde{\phi}_{k+\frac{2}{3}})\right)^\top \circ \left(\frac{\partial \Psi_{2k_0-2}^Y(ah)}{\partial \phi}(\phi_{k+1})\right)^\top$$

$$= \widehat{\widetilde{\Psi}^Y}_{2k_0-2}(ah) \circ \widehat{\widetilde{\Psi}^Y}_{2k_0-2}(bh) \circ \widehat{\widetilde{\Psi}^Y}_{2k_0-2}(ah),$$

where we used the notation

$$\tilde{\phi}_{k+\frac{2}{3}} = \Psi_{2k_0-2}^Y(-ah)(\phi_{k+1}), \quad \tilde{\phi}_{k+\frac{1}{3}} = \Psi_{2k_0-2}^Y(-bh) \circ \Psi_{2k_0-2}^Y(-ah)(\phi_{k+1}),$$

and $\widehat{\widetilde{\Psi}^Y}_{2k_0-2}$ the projection of the step backward associated to the state-adjoint system and $\Psi_{2k_0-2}^Y$ method. This completes the induction step and the proof. $\square$

In case of $k = 2$, Theorem B.2 in combination with equation 17 leads to the following expression

$$\widetilde{\Psi}_4^Y = W_{\frac{-(ah)^2}{16}}^{-1} \circ \Psi_{-ah}^{ALF2} \circ W_{\frac{a^2}{b^2}} \circ \Psi_{-bh}^{ALF2} \circ W_{\frac{b^2}{a^2}} \circ \Psi_{-ah}^{ALF2} \circ W_{\frac{-(ah)^2}{16}}.$$

The obtained results lead to the Algorithm 4 for the computation of gradients.

## C  DETAILS OF NUMERICAL EXPERIMENTS

In all the numerical experiments, our implementation of the Yoshida composition method uses the code of the MALI network (Zhuang et al., 2021). We use the steps forward and backward of the ALF method as composition steps to compute ALF2 and its Yoshida composition.

---

**Algorithm 4** Computation of gradients

---

1. Input: training data $z_0$, initialization of parameters $\theta$, velocity $v_0 = f(z_0, \theta_0)$
2. Propagate through the network using $\Psi_{2k}^Y$ to get $(z_N, v_N)$
3. Set $\lambda_N^z = \nabla L(z_N)$ and $\lambda_N^v = 0$
**for** k = N to 1 **do**
    4. Compute $\phi_k, \lambda_k$ from $\phi_{k+1}, \lambda_{k+1}$ using equation 17 and equation 22.
    5. Compute $\frac{\partial J(\theta)}{\partial \theta_k}$ using equation 11
**end for**
6. Output: gradients $\frac{\partial J(\theta)}{\partial \theta_k}$ for $k = 1, \ldots N - 1$.

---

## C.1 KEPLER PROBLEM

The training data for the comparison of the computational time in Table 2 is given by a trajectory $x$ of equation 12 with initial condition $x_0 = (0.75, 0, 0, \frac{0.9\pi}{4}\sqrt{\frac{5}{3}})$ on time interval $[0, T] = [0, 1]$, which is an elliptic orbit. The trajectory is obtained by numerical integration using `sci.integrate.odeint` with relative and absolute tolerances $10^{-7}$ and $10^{-8}$ respectively and maximum step size $10^{-5}$. The optimizer used in the training is SGD from PyTorch with initial learning rate 0.1 and scaled by 0.95 for each epoch. For completeness, we show the evaluation of the parameter error across the learning displayed as a function of time in Figure 2 and as a function of epochs in Figure 3. In the plots we show the results obtained with ALF, Y4 and also Runge-Kutta 4(5) (RK45), the latter is not a reversible method and requires storage of the intermediate states obtained during the integration forward. This implies additional memory consumption, namely, at each epoch the algorithm saves 8 additional states obtained during integration forward, making the memory consumption of the training higher. The four plots in Figures 3 and 2 are obtained for different initializations of the parameters $\alpha_0$ in the learning, namely, $\alpha_0 = 1.3, 0.1, 0.7, 0.75$.

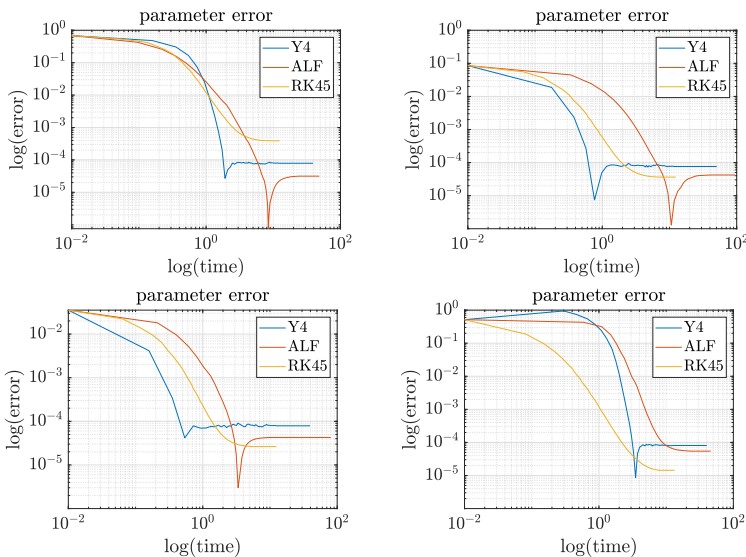

Figure 2: Error of the learned parameter with respect to the ground truth $\alpha$ as a function of time.

The error landscape in Figure 4 is obtained by considering 81 trajectories obtained using the same integration method as for the time comparison explained above for 81 different initial conditions $(x_0)_i$ in a neighborhood of $x_0$, given by a 4-dimensional box of diameter 0.4 around $x_0$. The points $(x_0)_i$ are chosen on a grid with a step size 0.1, which includes $x_0$ as its point. The behaviour of ALF and Y4 with adaptive stepping can be better understood when looking at fixed step methods, when the step size $h_i = h$ is fixed for all the steps. The loss landscape visualized in Figure 4 for fixed step ALF and fixed step Y4 shows that the minimum value of the loss is achieved at a better precision

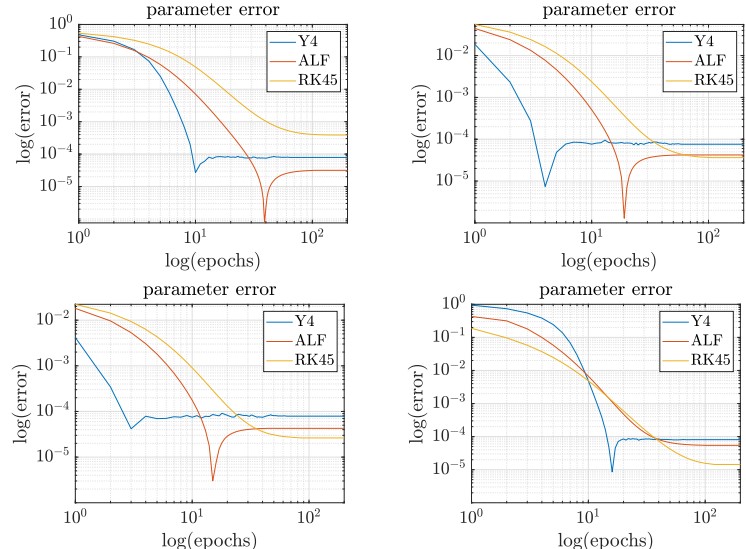

Figure 3: Error of the learned parameter with respect to the ground truth $\alpha$ as a function of epochs.

of the true parameter for the higher order methods than for the lower order method, which will be explained in more detail below. The loss visualized in Figure 4 as a function of $\alpha$ is

$$L(\alpha_k) = \frac{1}{81} \sum_{i=1}^{81} \sum_{j=1}^{5} \sum \|(q_{N_j}(\alpha_k))_i - q(t_j, (x_0)_i)\|^2$$

with $q_{N_j}$ projection of $z_{N_j}$ to $q$-coordinate and $\alpha_k$ taking 300 values in $[\frac{\pi}{4} - 10^{-4}, \frac{\pi}{4} + 10^{-4}]$. Here $(q_{N_j}(\alpha))_i$ is obtained by numerical integration of equation 12.

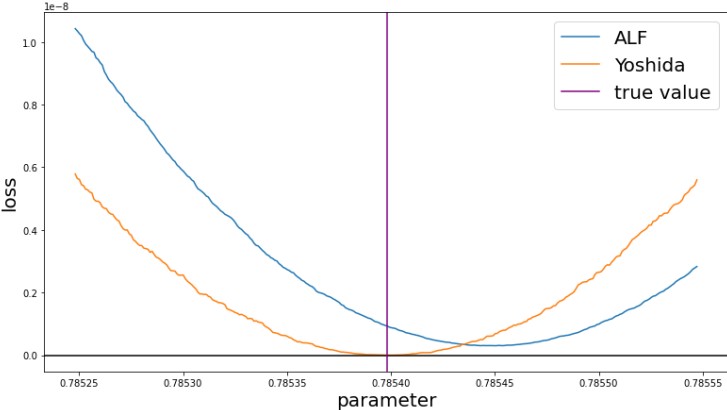

Figure 4: Error landscape of ALF and Y4 methods for Kepler problem showing the loss computed for the parameters in a neighbourhood of the true value of $\alpha$ displayed by a vertical line.

If $(q_{N_j}(\alpha))_i$ was obtained by exact integration of equation 12 and in the absence of noise and round-off errors, true parameter values constitute minima for $L$. We interpret the application of a numerical integrator as a perturbation of size $\mathcal{O}(h^p)$ to the exact $(q_{N_j}(\alpha))_i$, where $h$ is the step size of the integration and $p$ the order of the numerical method. This yields a perturbation $\tilde{L}$ of $L$ of size $\mathcal{O}(h^{2p})$ in case of the mean-square loss. Thus, assuming that the local minima of $L$ at the true parameter value is non-degenerate, $\tilde{L}$ has a local minimum within a ball around the true parameter of size $\mathcal{O}(h^p)$. This follows from classical discussions on the numerical conditioning of computing zeros of a function as, for instance, in (Dahmen & Reusken, 2022, §5.2). This provides a direct relation of the order of an integration method and the accuracy of identified parameters.

Notice that in the adaptive-step size context the perturbation of $L$ and, thus, the error of its minima are controlled by the provided error tolerance. However, the discussion shows that in order to be able to expect the same accuracy in the parameter identification, neural ODEs based on lower-order methods require more integration steps than neural ODEs based on high-order methods.

The above $\mathcal{O}(h^p)$ error relation in the parameter estimation constitutes an asymptotic upper bound. In Geometric Numerical Integration errors of numerical integrators can enter in highly symmetric way (Hairer et al., 2013). In symplectic integration of Hamiltonian systems, for instance, energy errors enter in an unbiased form. If the sought parameter is related to the geometric structure that is preserved by the geometric numerical integrator, parameters can potentially be estimated to higher accuracy than expected by the order of the numerical integrator. This, together with backward error analysis techniques, was used in (Offen & Ober-Blöbaum, 2022), for instance, to accurately identify a Hamiltonian function of a dynamical system even though a low order method was used to discretize the dynamical system. These techniques, however, are tailored to the geometric problem at hand, while the approach of this article considers a more general case.

### C.2 NONLINEAR HARMONIC OSCILLATOR

There are two settings considered for the learning of the dynamics equation 13. In the first setting, the learning problem is the parameter identification as presented in Section 4.1.2. In the second case we consider the parametrization of the potential by a neural network as described in Section 4.2.1. Here we give more details on both problems.

#### C.2.1 IDENTIFICATION OF PARAMETERS

In the experiments for the time comparison shown in Table 3 we consider a set of 200 trajectories in the training data with the initial conditions generated by the Halton sequence in a 20-dimensional box around zero vector $x_0$ with diameter 2.0. The trajectories are obtained by numerical integration using `sci.integrate.odeint` with relative and absolute tolerances $10^{-13}$ and $10^{-14}$ respectively. In the training we use `AdamW` optimizer from `PyTorch` with learning rate scheduler `ExponentialLR`. The results shown in Table 3 are obtained with different learning rates, namely, the first two with the initial learning rate $10^{-2}$ and $\gamma = 0.995$, the last three with the initial learning rate $10^{-1}$ and $\gamma = 0.998, 0.997, 0.99$ for the tree results respectively. At each epoch we consider all 200 trajectories, so that the loss is $L = \frac{1}{200} \sum_{i=1}^{200} \|(z_N)_i - x(T, (x_0)_i)\|^2$ with $T = 0.5$. While Table 3 compares the training time of ALF and Y4, it is also important to compare their performance in the learned parameters. In Figure 5 we show the results in the error of the learned parameters as a function of computational time measured at each epoch of ALF, Y4 and also RK45, which is not reversible. We can see that Y4 in not only faster than ALF in the training but the same also holds for the error in the learned parameters. While RK45 is the fastest to get to accurate parameters, it also requires the storing of 80 additional states during integration at each epoch, which means a considerable contribution to the memory costs. To better understand the reasons of the faster learning of Y4 than ALF, we show in Figure 6 the computation time accumulated at each epoch of the training. The computational time per epoch is smaller for Y4, which contributes to the faster convergence in the training. In both Figures 5 and 6, the four plots correspond to different random initializations of the parameters in the optimization.

In addition to results obtained for adaptive stepping, we test ALF and Y4 with the step size fixed to $h = 0.1$ and the training until either the training accuracy reaches $10^{-4}$ or the number of epochs reaches 500. Figure 7 shows that ALF is stuck at the training accuracy $10^{-2}$ and the training stops because of reaching 500 epochs, while Y4 converges to accuracy $10^{-4}$ with 181 epochs. The same behaviour is observed for different parameter initialization. Decreasing the step size to $h = 0.01$ permits ALF to reach accuracy $10^{-4}$. The results obtained in Figure 7 show that with a fixed step size the lower order method is unable to achieve an accuracy better than $10^{-2}$ in training loss, whereas Y4 reaches accuracy $10^{-4}$. This illustrates what happens in the case of the adaptive time-stepping. A lower order method needs to reduce the step size to get to better accuracy. This implies more steps in the integration, and therefore, slower computations.

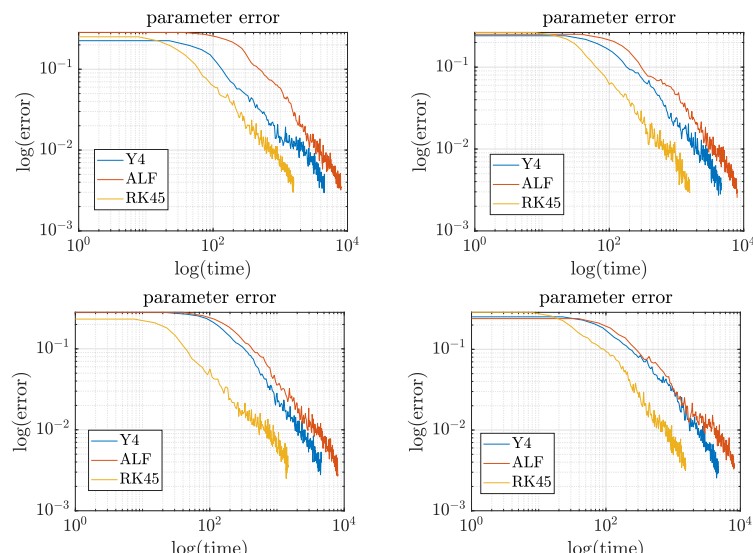

Figure 5: Error in learned parameters as a function of time.

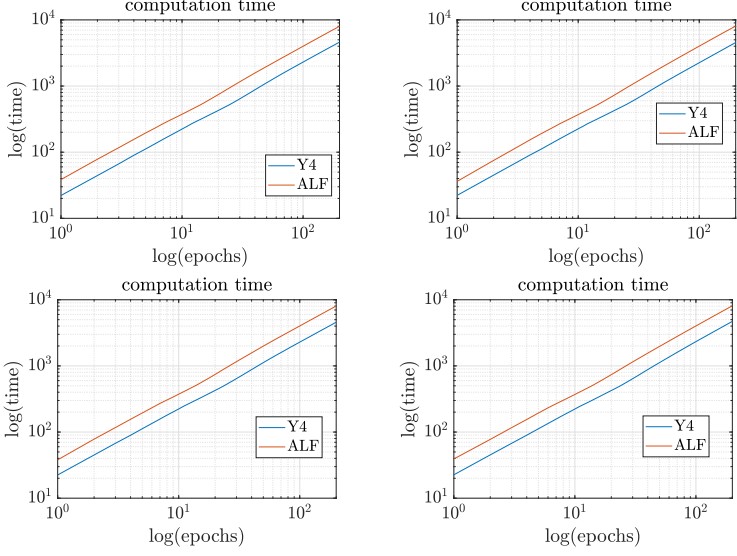

Figure 6: Time of computation in function of epochs. When the curve is positioned lower, the corresponding algorithm is faster.

### C.2.2 NEURAL NETWORK PARAMETRIZATION

The goal is to find the unknown potential governing equation 14. For this we assume a particular form of the potential, namely,

$$V(q) = \sum_{i=1}^{s} \sum_{j=1}^{n} c_{i,j} \sigma_i(q_j) + \sum_{i=1}^{d} \sum_{j=1}^{n} \sum_{k=j+1}^{n} C_{i,j,k} \Sigma_i(\|q_j - q_k\|),$$

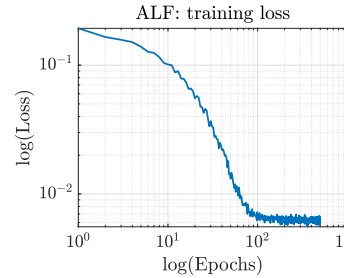 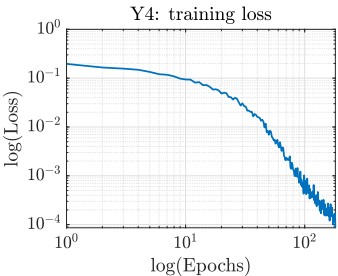

Figure 7: Training loss is displayed in logarithmic scale for the parameter identification in case of coupled oscillation for ALF and Y4 with fixed step size $h = 0.1$.

where $\sigma_i$ stand for different single particle potentials and $\Sigma_i$ for double particle potentials. In the case considered above, we have

$$c_{1,1} = \frac{a_1}{2}, \; c_{1,2} = \frac{a_2}{2}, \; \sigma_1(q) = q^2,$$

$$c_{2,1} = \frac{b_1}{4}, \; c_{2,2} = \frac{b_2}{4}, \; \sigma_2(q) = q^4,$$

$$C_{1,1,2} = \frac{e}{2}, \; \Sigma_1(x) = x^2.$$

In the learning problem, we assume that functions $\sigma_1, \sigma_2$ and $\Sigma_1$ are unknown as well as parameters $a_1, a_2, b_1, b_2, e$. We parameterize the derivatives $\frac{a_1}{2}\sigma_1', \frac{a_2}{2}\sigma_1', \frac{b_1}{4}\sigma_2', \frac{b_2}{4}\sigma_2'$ and $\frac{e}{2}\Sigma_1'$ by neural networks each and use them to model the dynamics in equation 14. We use 5 neural networks, which we denote by $\xi_1, \xi_2, \xi_3, \xi_4, \xi_5$. All of them have the same architecture $q \mapsto W_1 \tanh(W_2 \tanh(W_3 q))$, where $W_1$ is a matrix of parameters of size $1 \times 100$, matrix $W_2$ is of size $100 \times 100$ and $W_3$ is of size $100 \times 1$. The resulting dynamics is defined by

$$\dot{q}_1 = v_1, \quad \dot{v}_1 = -\xi_1(q_1) - \xi_3(q_1) - \xi_5(q_1 - q_2),$$
$$\dot{q}_2 = v_2, \quad \dot{v}_2 = -\xi_2(q_2) - \xi_4(q_2) - \xi_5(q_2 - q_1).$$

The equations parameterized by neural networks are then integrated using ALF or Yoshida composition of ALF2 at each epoch in the training. The training data is set to be a set of 1000 trajectories with the initial conditions generated by the Halton sequence in a 4-dimensional box around $x_0 = (0.8, -0.4, 0.0, 0.0)$ with diameter 2.0. The optimizer is `AdamW` with initial learning rate $10^{-3}$ and scheduler `ExponentialLR` with $\gamma = 0.995$. In addition, we consider batches of 300 trajectories at each epoch with the resulting loss function of the same form as in the case of the parameter identification problem.

## C.3 DISCRETIZED WAVE EQUATIONS

For generation of the training data, we consider the wave equation with potential $V(u) = \frac{1}{2}u^2$. The true motions can be expressed in the time-dependent Fourier series as

$$u(t, x) = \sum_{m=-\infty}^{\infty} \hat{u}_m(t) e^{2\pi i m x/L}, \quad L = 1$$

where the Fourier coefficients evolve as

$$\hat{u}_m(t) = \gamma_m^{-1} \hat{v}_{m,0} \sin(\gamma_m t) + \hat{u}_{m,0} \cos(\gamma_m t), \quad \gamma_m = \sqrt{1 + \frac{4\pi^2}{L^2} m^2}.$$

Here $\hat{u}_{m,0}$, $\hat{v}_{m,0}$ are the Fourier coefficients of an initial wave $u(0, x)$ and velocity $u_t(0, x)$, respectively. Notice that a Fourier coefficient $\hat{u}_m(t)$ remains exactly zero over time if and only if $\hat{u}_{m,0} = 0 = \hat{v}_{m,0}$. Training data to initial data with only finitely many nonzero Fourier coefficients can, therefore, be obtained to machine precision by a spectral method. Alternatively, solutions can be computed by an application of the 5-point stencil as described in Example 7 (16) in (Offen & Ober-Blöbaum, 2024) on a fine mesh with discretization parameters $\Delta t = 1/160$, $\Delta x = 1/80$ and then subsampled to a mesh with $\Delta t = 1/40$, $\Delta x = 1/20$. In our case both methods yield the same

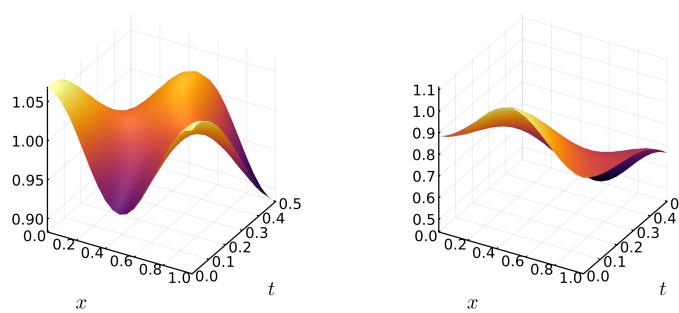

Figure 8: Two samples of the training data used in section C.3.

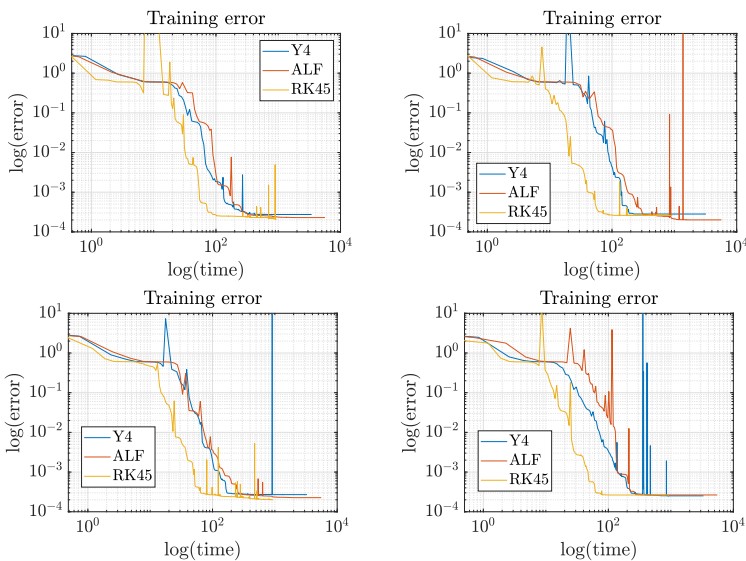

Figure 9: Time of computation in function of epochs. Lower curve means faster computations.

training data up to a maximum error of order $1\mathrm{e} - 4$. In the training data creation, we sample initial $\hat{u}_{m,0}$, $\hat{v}_{m,0}$ from a standard normal distribution. It is then weighted by $e^{-4m^8}$ such that effectively only the first two Fourier modes are active. See figure 8 for a plot of two of the solutions to the wave equation that were used to create the training data set. In the training, we consider initial and final points of 50 trajectories on time interval $[0, 0.3]$ and 30 unseen trajectories in the testing. We use the optimiser LBFGS with the default values of the parameters. In the numerical tests, we compare the behaviour of ALF, Y4 and Runge-Kutta 4(5). It can be seen in Figure 9 that Y4 reaches the lowest values in the training loss faster than ALF. While RK45 is fastest, it also consumes more memory, which can make a crucial difference in high dimensional systems. We also report a lower time of computations per epoch for Y4 with respect to the results by ALF in Figure 10.

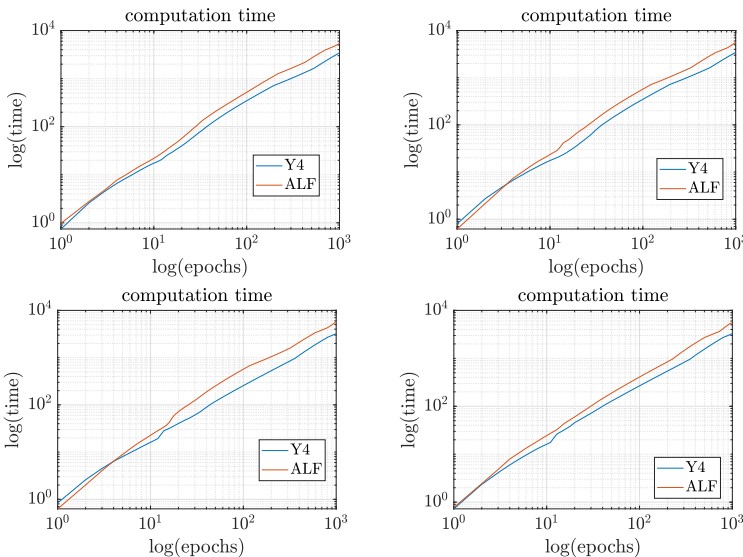

Figure 10: Time of computation in function of epochs. When the curve is positioned lower, the corresponding algorithm is faster.

