# OpenReview forum: "Adaptive higher order reversible integrators for memory efficient deep learning"
_ICLR.cc/2025/Conference — Submitted to ICLR 2025_

### Official Review · Reviewer_zgYE · 2024-10-22

**Soundness:** 2
**Presentation:** 4
**Contribution:** 3
**Rating:** 6
**Confidence:** 4

**Summary:**

In summary, the paper presents a novel approach to construct higher-order reversible integrators for memory-efficient deep learning, with the main contribution being the development of higher-order reversible integrators based on the asynchronous leapfrog (ALF) method. The proposed methods demonstrate improvements in training speed and accuracy, particularly in learning dynamical systems. It opens up new avenues for training neural ODEs with restricted memory budgets.

**Strengths:**

- I find the idea novel and original. The paper extends the capabilities of existing reversible integrators like ALF and reversible Heun, which were previously limited to low orders of accuracy. By demonstrating the feasibility of higher-order reversible integrators, the authors open new possibilities for improving the efficiency and accuracy of neural ODE-based deep learning.

- The paper is presented with exceptional clarity, guiding readers seamlessly through key concepts. It begins with a thorough setup of neural ODEs, followed by a well-balanced discussion of the advantages and limitations of two gradient computation methods, effectively motivating the notion of reversibility. The structured flow continues with an in-depth explanation of numerical methods, culminating in the core contribution: the construction of higher-order methods using ALF or Heun’s method. The integration of these methods within the neural ODE training framework (i.e. leveraging state-adjoint equations), is explained with precision, enhancing comprehension and practical relevance.

- The significance of this work is moderate. While it can be seen as a direct follow-up to the contributions of Zhuang et al. (2021) and Kidger et al. (2021), it still brings meaningful value, particularly in the context of neural network training for dynamical systems with multiple time scales.

**Weaknesses:**

- My primary concern lies with the soundness of the conclusion. Indeed, we see smaller errors and computation time of adaptive Y4 when comparing with adaptive ALF in both Kepler and nonlinear harmonic oscillator under some settings. It's not clear to me:
   -  Where the savings of the computation time come from? Is it because adaptive Y^4 in general uses larger step sizes? Is there a limit on $k$ for adaptive $Y^{2k}$ to outperform adaptive ALF on computation time? Why or why not there is a limit? I would suggest the authors to add more comparisons by varying parameter $k$ and report the results.
   -  Why the final loss of Y4 is lower? (When solving differential equations, it's easier to understand that higher order methods often outperform lower order methods w/ the same step size. However, when training neural networks, it's less obvious - the training is more or less "agnostic" to the computation schemes you are incorporating ... eg. why Runge Kutta is better than Euler?) What if we increase the number of epochs? Will it help adaptive ALF to achieve lower loss? Could the authors provide some justifications and experimental data to support them?

- The author may consider adding more contexts for the "adaptive stepping" presented in Hairer et al. (2006) in the Background section.

- The authors may consider adding a broader set of literatures for the readers. For examples,
  - known reversible methods other than asynchronous leapfrog and Heun:
     - Loup Verlet. Computer” experiments” on classical fluids. i. Thermodynamical properties of Lennard-Jones molecules. Physical review, 159(1):98, 1967.
  - for composition of mixed-order methods, although it is less developed than the theory for composing same-order integrators, there are some frameworks addressing this topic, especially in contexts like symplectic integrators and splitting methods
     - "Splitting methods for differential equations" by Sergio Blanes, Fernando Casas, Ander Murua
     - “Geometric Numerical Integration” by Hairer, Lubich, and Wanner
  - some other works on deep learning of dynamical systems
     - Rudy SH, Kutz JN, Brunton SL. 2019 Deep learning of dynamics and signal-noise decomposition with time-stepping constraints. J. Comput. Phys. 396, 483–506. (doi:10.1016/j.jcp.2019.06.056)
     - Y. Liu, J. N. Kutz, and S. L. Brunton, “Hierarchical deep learning of multiscale diﬀerential equation time-steppers,” arXiv:2008.09768
(2019)
     - Raissi M, Perdikaris P, Karniadakis GE. 2018 Multistep neural networks for data-driven discovery of nonlinear dynamical systems. Preprint. (https://arxiv.org/abs/1801.01236)

**Questions:**

I don't quite understand the purpose of section 4.2. "LEARNING OF DYNAMICAL SYSTEMS PARAMETERIZED BY NEURAL NETWORK". Could the authors comment on the novel insights or advantages gained from this section compared to 4.1, or to clarify how this section extends or complements the results from 4.1?

---

> ### Author Response · Authors · 2024-11-27
>
> Dear reviewer, we are indebted for your careful reading and helpful comments. We would like to address your questions and concerns.
>
>
> 1. *Where the savings of the computation time come from? Is it because adaptive $Y^4$ in general uses larger step sizes? Is there a limit on $k$ for adaptive $Y^{2k}$ to outperform adaptive ALF on computation time?*
>
> In the considered examples, the 4th order integrator $Y^4$ and the (effectively) second order integrator ALF with adaptive time-stepping are applied to differential equations that resemble the planar Kepler problem and a nonlinear coupled Duffing harmonic oscillator with 10 masses. These are integrated forward (with adaptive step size control) and the adjoint equations are integrated backwards with the step sizes from the forward integration repeatedly. Indeed $Y^4$ can meet the local error tolerance with fewer steps than ALF as it is of higher order. This causes computational savings for the forward and the backward pass. This is because in this regime of local error tolerance and on the differential equation at hand, a fourth order method turns out to be more efficient. Indeed, higher order methods can be considered. Eventually, we expect a limit in efficiency gain just as in classical integration methods where there exists an optimal order. This is because if $c_p h^{p+1}$ is a bound for the local error of a numerical integrator that holds for all step sizes $0<h<h_{\mathrm {max}}$ then the error constant $c_p$ tends to rise with the order $p$ of a method, as is known in classical numerical integration theory. Indeed, these error constants grow when higher order methods $Y^{2k}$ are constructed as Yoshida compositions.
>
> 2. *Why the final loss of Y4 is lower? What if we increase the number of epochs? Will it help adaptive ALF to achieve lower loss?*
>
> We see that the prominent display of Figure 3 might be misleading as it is irrelevant for the hypothesis of the article. It has, therefore, been moved to the appendix (see Figure 7). Indeed, we do not see any reason to expect Y4 to be better trainable than ALF-based networks *per epoch*. The article's hypothesis is that computational advantages arise because an epoch can be computed faster thanks to higher order integration. Moreover, in our newly introduced example (see C.3 Discretized wave equation) the training accuracy after a sufficiently large training time levels out at roughly the same value.
>
> 3. *The author may consider adding more contexts for the "adaptive stepping" presented in Hairer et al. (2006) in the Background section.*
>
> We have added a paragraph in the introduction, which explains the concept of adaptive step size selection and provides references.
>
> 4. *The authors may consider adding known reversible methods other than asynchronous leapfrog and Heun*
>
> We have added this classical reference, on which the ALF method is based.
>
> 5. *for composition of mixed-order methods, although it is less developed than the theory for composing same-order integrators, there are some frameworks addressing this topic, especially in contexts like symplectic integrators and splitting methods*
>
> We have noticed that our terminology *mixed-order* could be mistaken for compositions of integrators of mixed orders, while in the context of the article *mixed-order* refers to the fact that a single step of ALF in its augmented phase space yields a prediction of the state variable to second order but a prediction of a velocity variable only to first order in the step-size. We have adapted our formulations and avoid the term *mixed-order*. Moreover, we have included the suggested excellent references.
>
>
> 6. *some other works on deep learning of dynamical systems*
>
> The suggested references are now included in the introduction.
>
> 7. *I don't quite understand the purpose of section 4.2. "LEARNING OF DYNAMICAL SYSTEMS PARAMETERIZED BY NEURAL NETWORK". Could the authors comment on the novel insights or advantages gained from this section compared to 4.1, or to clarify how this section extends or complements the results from 4.1?*
>
> Indeed, the additional experiment supports the same messages as the experiments of section 4.1. It is included to demonstrate that our method can be applied to problems with many parameters. Moreover, an additional experiment was included (discretized wave equation) to show the applicability to high-dimensional dynamical systems.

---

### Official Review · Reviewer_6eWA · 2024-10-27

**Soundness:** 3
**Presentation:** 2
**Contribution:** 2
**Rating:** 5
**Confidence:** 4

**Summary:**

"Improving neural network architectures inspired by ODEs" and "using neural networks to learn ODEs" have been investigated alternately. This paper proposes a novel architecture based on a composition method, which improves model design and achieves memory efficiency during training thanks to its reversibility.

**Strengths:**

The focus of the study is reasonable, with a thorough survey of related works and sound theoretical foundations. If this paper were a contribution to applied mathematics, it would deserve high praise.

**Weaknesses:**

However, the validation falls short. The experiments are limited to simple datasets, which are insufficient to demonstrate the general applicability of the proposed architecture. In what settings is this architecture genuinely effective? For instance, while reversible dynamical systems, like the Kepler problem used in the study, are a clear application, this alone is somewhat narrow, as one could achieve similar outcomes by applying reversible integrators within Neural ODE frameworks. On the other hand, could applying this architecture to general tasks, like image processing, compromise performance due to the reversibility constraint? The study would benefit from a clarified scope of applicability and a more comprehensive set of experiments.

While the theoretical memory efficiency is convincing, there is no experimental validation. Although ALF is presumably the baseline, only computation time is compared. An experimental comparison of memory consumption is also necessary, along with theoretical orders for computational and memory costs.

Considering ICLR's focus on machine learning, this paper falls short of expected standards due to the ambiguity of its contributions and insufficient empirical support.

**Questions:**

See weakness.

Minor comments:
- References are cited without parentheses, which hinders readability. Proper citation using \citep tags with parentheses is recommended.
- The composition method could construct more complex architectures, and such an exploration could add value to the study.

---

> ### Author Response · Authors · 2024-11-27
>
> Dear reviewer, we highly appreciate your helpful comments and would like to address your questions and concerns.
>
> 1. *The validation falls short. The experiments are limited to simple datasets, which are insufficient to demonstrate the general applicability of the proposed architecture. In what settings is this architecture genuinely effective? For instance, while reversible dynamical systems, like the Kepler problem used in the study, are a clear application, this alone is somewhat narrow, as one could achieve similar outcomes by applying reversible integrators within Neural ODE frameworks. On the other hand, could applying this architecture to general tasks, like image processing, compromise performance due to the reversibility constraint? The study would benefit from a clarified scope of applicability and a more comprehensive set of experiments.*
>
> We have clarified the scope of the method in the introduction, which we mainly see in the identification of high-dimensional dynamical systems from irregular time-series data, for which gradient-based optimization methods can cause high memory requirements. This is now reflected better in the conclusion as well. To show the applicability of our method to such systems, we have included the example of learning a dynamical system which is governed by a spatially discretized partial differential equation. This is a highly active research area and references to alternative approaches based on model-order reduction are provided. Indeed, an application of high order reversible Neural ODE architectures to image processing tasks and normalizing flows is an exciting field for exploration in future works.
>
> 2. *While the theoretical memory efficiency is convincing, there is no experimental validation. Although ALF is presumably the baseline, only computation time is compared. An experimental comparison of memory consumption is also necessary, along with theoretical orders for computational and memory costs.*
>
> The memory requirements of the methods discussed in the article are now compared in Table 1. Experimental validation of lower memory costs for neural ODEs that are based on reversible methods in comparison to neural ODEs based on non-reversible methods that require storage of computational graphs has been provided in [1]
>
> 3. *Considering ICLR's focus on machine learning, this paper falls short of expected standards due to the ambiguity of its contributions and insufficient empirical support.*
>
> In the revised version, we state the contribution to this field more clearly. Moreover, the new example of learning a discretized partial differential equation constitute additional empirical support. The scope of the paper, learning of models of dynamical systems from data, is a highly active topic in machine learning and contributions are regularly presented at machine learning conferences. Examples include HNN [2] (NeurIPs 2019), LNN [3] (ICLR), MALI [1] (ICLR2021), or [4,5]({AAAI} 2020/2022 Symposium on Physics Guided AI).
>
> 4. *References are cited without parentheses, which hinders readability. Proper citation using citep tags with parentheses is recommended.*
>
> The citation has been improved.
>
> 5. *The composition method could construct more complex architectures, and such an exploration could add value to the study.*
>
> Indeed, it is an exciting avenue to explore higher order compositions or compositions which have at a given order an optimal error constant. Additionally, symmetry properties can be explored in combination with our reversible, time-adaptive approach. In order not to diffuse the main messages of the contribution and align with the conference's focus, we have decided not to include content of this context.
>
> **References**
>
> [1] Juntang Zhuang, Nicha C. Dvornek, Sekhar Tatikonda, and James S. Duncan. Mali: A memory efficient and
> reverse accurate integrator for neural odes, 2021
>
> [2] Samuel Greydanus, Misko Dzamba, and Jason Yosinski. Hamiltonian Neural Networks. Advances in Neural Information Processing Systems, 2019.
>
> [3] Miles Cranmer, Sam Greydanus, Stephan Hoyer, Peter Battaglia, David Spergel, and Shirley Ho. Lagrangian
> neural networks, 2020.
>
> [4] Christine Allen-Blanchette, Sushant Veer, Anirudha Majumdar, and Naomi Ehrich Leonard. LagNetViP: A
> Lagrangian neural network for video prediction (AAAI 2020 symposium on physics guided ai), 2020.
>
> [5] Justice Mason, Christine Allen-Blanchette, Nicholas Zolman, Elizabeth Davison, and Naomi Leonard. Learning
> interpretable dynamics from images of a freely rotating 3d rigid body, 2022.

---

> > ### Comment · Reviewer_6eWA · 2024-11-28
> >
> > Dear Authors,
> >
> > Thank you for your detailed response.
> >
> > > We have clarified the scope of the method in the introduction.
> >
> > I understand that the primary objective of your work is "the identification of high-dimensional dynamical systems from irregular time-series data." However, considering that comparison methods such as ACA and MALI also address applications in image processing, the scope of your method seems somewhat narrow. Could you clarify why your method is not suitable for image processing? Alternatively, could you specify the advantages that make your method particularly well-suited for dynamical systems?
> >
> > I appreciate the revisions you have made to the manuscript, which have clarified the scope of your work. Nevertheless, I have the following remaining concerns:
> >
> > 1. In Tables 2-5, the "asynchronous ALF" appears to differ slightly from MALI. If so, this means that a direct comparison with existing methods has not been performed. Even if "asynchronous ALF" is identical to MALI, I believe it is still necessary to include comparisons with general NODEs and ACA, as summarized in Table 1. (RK45 in Figure 5 corresponds to NODEs?)
> >
> > 2. Instead of simply listing numerical results in tables, it would be more informative to include visual representations of the data, such as graphs showing means, variances, and temporal changes. Figures 5-6 also appear to display all trials; would it not be clearer to present means with error bars?
> >
> > 3. Intuitively, using a reversible integrator for irreversible dynamical systems does not seem appropriate. The wave equation in the added experiments is also a conservative system. Could you experimentally and theoretically demonstrate the applicability of your method to dissipative systems as well?
> >
> > Thank you again for your efforts in addressing these points.
> >
> > Sincerely,

---

> > > ### Author Response · Authors · 2024-11-28
> > >
> > > Dear Reviewer,
> > >
> > > Thank you for swift reply and your additional questions and comments.
> > >
> > > **Applicability to image processing:** Indeed, the proposed high order integrator can be used in NODEs to model normalizing flows, for instance, or other image processing tasks, just as MALI. However, we expect to see the advantage of higher order reversible integration in NODEs over low order integration mostly in systems in which parameters need to be identified to very high accuracy. An exciting avenue to explore is where such an advantage can give an edge over existing methods in image processing applications. Potential applications in image processing could include the generation of video data of dynamical systems [1,2], when these need to show physically correct behaviour to very high accuracy.
> > >
> > > 1. NODE is a class of networks based on the adjoint method for gradient computation and can be combined with any numerical integration method. Both MALI and ACA have the structure of NODE, where MALI is a NODE based on ALF and ACA can be based on any numerical integrator and its additional feature is the adaptive checkpointing. As MALI is based on ALF and our proposed network has the same NODE structure but is based on Yoshida compositions of ALF, we refer to the networks by the names of the corresponding integrators. We present the comparison of the proposed method with MALI, because MALI is the optimal network in the class of networks with the NODE structure in terms of the memory cost and gradient accuracy and is the baseline for the comparison. Extensive numerical comparisons of MALI with other neural network architectures are included in the original MALI paper. As our proposed method can be interpreted as a higher order version of MALI, we concentrate or numerical experiments supporting the hypothesis that reversible, higher order integration can pay off for NODEs. We have also included ACA based on RK45 to serve as a comparison with a NODE combined with a standard integrator.
> > >
> > > 2. The tables and graphs provide empirical evidence supporting the theoretical argumentation of the article. While there are various ways of presenting these, we think that our form of presentation does not hinder interpretability or understanding. We would have included the visual representations of the results suggested be the reviewer, if the editing window was not over.
> > >
> > > 3. In NODEs a parametrized differential equation is integrated by a numerical integration method. In this context, "reversible" means "easy to invert". The "easy to invert"-feature is what matters in the reversible NODE setting: during training the adjoint system has to be integrated backwards in time. This needs to be done with the inverse of the numerical method in order to compute exact gradients. There is another, related notion of "reversible": In numerical analysis (NA) a numerical integrator is "reversible" when its inverse can be obtained by a simple change of sign of the step size. When a numerical integrator is "reversible" in the sense of NA and it is explicit, then it is also "reversible" in the NODE sense as it is easy to invert: just flip the sign of the step-size. This applies to the ALF method and its Yoshida composition that we are proposing.
> > > These two notions of "reversible" may not be confused with geometric properties of a differential equations or an integrator, such as "time-reversibility" in classical mechanics. While reversible NODEs can be applied to systems with dissipation or dispersion, the adjoint system backward in time is expanding. This can lead to instabilities in the computation of gradients for training that need to be overcome either with checkpointing or with small integration steps. We, therefore, expect the clearest advantages of high order reversible NODEs in non-dissipative systems. Even a forward pass can be problematic as the considered Yoshida compositions make use of negative time-step sizes. These limitations are well understoond in the context of composition methods, see, for instance, [3].
> > >
> > > **References**
> > >
> > > [1] Christine Allen-Blanchette, Sushant Veer, Anirudha Majumdar, and Naomi Ehrich Leonard. LagNetViP: A Lagrangian neural network for video prediction (AAAI 2020 symposium on physics guided ai), 2020.
> > >
> > > [2] Justice Mason, Christine Allen-Blanchette, Nicholas Zolman, Elizabeth Davison, and Naomi Leonard. Learning interpretable dynamics from images of a freely rotating 3d rigid body, 2022.
> > >
> > > [3]  Hairer, Lubich, Wanner:  Geometric Numerical Integration: Structure-Preserving Algorithms for Ordinary Differential Equations, Springer (2006)

---

> > > > ### Comment · Reviewer_6eWA · 2024-11-29
> > > >
> > > > Dear Authors,
> > > >
> > > > Thank you very much for your detailed response. However, I believe my wording might not have been clear enough, and my intentions may not have been fully conveyed.
> > > >
> > > > >These two notions of "reversible" may not be confused with geometric properties of a differential equations or an integrator, such as "time-reversibility" in classical mechanics.
> > > >
> > > > You mentioned that this method is intended for "the identification of high-dimensional dynamical systems from irregular time-series data," correct? In that case, the time-reversibility of the target dynamical systems should be important, I think. This is because reversible integrators are specifically designed to accurately reproduce the properties of time-reversible dynamical systems. This point is also discussed in Chapter V of the reference [3] you cited.
> > > >
> > > > Thus, applying a reversible integrator to time-irreversible dynamical systems may lead to unforeseen issues. To demonstrate the absence of such problems, I believe it would be better to provide a theoretical analysis or experiments on large or complex time-irreversible (i.e., dissipative) systems.
> > > >
> > > > I am afraid that such issues have not surfaced simply because the dynamical systems examined in Section 2 are all time-reversible.

---

> > > > > ### Author Response · Authors · 2024-11-29
> > > > >
> > > > > Dear Reviewer,
> > > > >
> > > > > Thank you for clarifying your concerns. Our approach is based on the composition of ALF steps, therefore, we can rely on the results in MALI paper, which provides experiments with dissipative systems [1]. Their experiments demonstrate that the favourable behaviour of reversible NODEs does not depend on special geometric properties of the differential equation.
> > > > >
> > > > > [1] Juntang Zhuang, Nicha C. Dvornek, Sekhar Tatikonda, and James S. Duncan. Mali: A memory efficient and reverse accurate integrator for neural odes, 2021

---

> > > > > > ### Comment · Reviewer_6eWA · 2024-12-01
> > > > > >
> > > > > > Dear Authors,
> > > > > >
> > > > > > In my understanding, the MALI paper examined only one dynamical system, the Mujoco time-series, using a latent-ODE, which is a very small-scale task and is limited as a demonstration. If this method is designed for dynamical systems, it is necessary to discuss the distinction between reversible and irreversible dynamics with technical rigor and dedicated experiments, as many researchers have devoted considerable efforts to this area.
> > > > > >
> > > > > > For instance, Section XII in the book [3] discussed the impact of dissipative perturbations on Hamiltonian systems. Moreover, the GENERIC formulation was proposed to handle the combination of reversible and irreversible dynamics, and the GENERIC integrator was specifically proposed for such formulation. These facts suggest that simply applying a reversible integrator to irreversible dynamics may be inadequate.
> > > > > >
> > > > > > This paper does not discuss this aspect sufficiently and does not verify the generality of the proposed method theoretically or empirically. Hence, it is difficult to value this work highly in its current form.

---

> > > > > > > ### Author Response · Authors · 2024-12-02
> > > > > > >
> > > > > > > Dear Reviewer,
> > > > > > >
> > > > > > > The article's goal is to introduce higher order integrators for neural ODEs, with the same favourable memory costs and flexibility as the ALF method in the MALI-article. As our proposed method can be interpreted as a higher order version of the ICLR contribution MALI, the main claim of the article is that higher order integration can pay off, which we verify in various examples including high-dimensional cases. The concerns raised in your comments, however, are directed towards the general applicability of neural ODEs that are based on reversible integrators, which is not the focus of our contribution. Moreover, any such experiments cannot be provided at this stage of the review process.

---

### Official Review · Reviewer_dws7 · 2024-11-04

**Soundness:** 2
**Presentation:** 2
**Contribution:** 1
**Rating:** 3
**Confidence:** 4

**Summary:**

This paper uses time-reversible ODE integration to solve neural ODEs. It focuses on a specific class of higher-order reversible integrators, and uses the time reversibility property to avoid inconsistency between the system states $z$ on the forward vs. backward passes. This allows a backward propagation of gradient information with memory costs independent of the sequence length.

The authors demonstrate this strategy for tuning parameters of dynamical systems or learning ODE terms parametrized by neural networks.

**Strengths:**

The idea of using higher-order reversible integrators to solve ODEs is inherently appealing, as the potential to reduce memory costs and improve accuracy is good. The ideas are mostly clearly explained, and the examples, if somewhat simplistic, are at least relevant to the stated goals.

**Weaknesses:**

Overall the paper doesn't provide significant theoretical results, and the experimental results are fairly weak (details below):

Literature review and citations are inadequate. There is a considerable and active literature on learning dynamical systems with ML, but this hasn't really been acknowledged. As a start, foundational papers such as Chen (2018) and Rackauckas (2020) should be cited, and many of the references in Ghadami (2022) will be relevant. There is also a considerable literature on physics-constrained (e.g. McGreivy 2023) and Hamiltonian neural networks (Greydanus 2019 and later articles) that has not been adequately cited. The first paragraph of the introduction makes fairly broad claims ("learning models of dynamical systems") with loose terminology ("numerical methods") but the cited references are all highly specific, a broad overview of the field should be given with major relevant works cited.

The assessments of improvements to speed/accuracy tradeoffs arising from the proposed approached are quite weak:
* In both tables 2 and 3 we never compare to alternative methods or even weak baselines such as a non-reversible integrator with adaptive step size (e.g. Forward Euler or RK4), but rather only between low- and high-order version of reversible integrators. We are missing a clear picture of how the proposed technique improves on alternative methods. It can' simply be that higher-order integrators are more efficient on these particular problems, that's simply not of sufficient general interest
* For parameter estimation tasks, we should consider accuracy with regard to the reference (ground truth) parameters, or prediction errors on held-out initial conditions. The training loss doesn't necessarily indicate which method is actually better in these senses.
* A fixed step size was used in table 2, providing an unrealistic picture of how this would be used and losing sight of the fundamental reasoning behind the technique
* Fig. 2 seems to exaggerate the benefits of Y4 over ALF, since in a realistic scenario the step size would be automatically set by each method. The loss landscape should (also) be shown for adaptive step sizes scenario, and for the same 10^-8 accuracy tolerance as used for computing table 1.
* The problems are fairly easy. The system state was fairly low dimensional in all examples. The only example where a neural network was trained involved learning a low-dimensional potential with strong inductive biases from physics. There are many examples of challenging dynamical system learning tasks in the literature, including some benchmark studies. Ideally we should see an improvement of cost-accuracy tradeoffs on some established task.

One of the advantages of backward mode differentiation over forward mode is that we don't need to compute input-output gradients $d\phi_{j+1}/d\phi_j$ which are of size $n^2$. But here we do, at least in the current formulation, calculate and store the matrix explicitly. Is this strictly necessary, or is there some way around it? What happens when $d\phi_{j+1}/d\phi_j$ is sparse, for example when representing local interactions in a discretized PDE? The authors claim that their time integrators could work for "potentially very high-dimensional $z$" (line 173) but this doesn't seem to scale.

Checkpointing for gradient calculation should be mentioned as an alternative to the present approach for reducing memory costs, and the relative merits of each should be discussed.

For the proposed tasks and dataset sizes, many other approaches are possible. Examples include simulation-based inference (Cranmer 2020) that treats the forward process as a black box, or simply training an attention-based or recurrent network to impute missing in continuous time. These should at least be discussed, with references.

The notation in 2.1 is fairly confused. $\sigma$ is supposed to be an activation function but then turns out to have learnable parameters. The role of $h$ in eq. 3 isn't clearly stated. $F$ seems to be used once for $f$, and $T$ is used without definition. At the same time, there is probably too much detail here, as we don't really need an explanation of backprop.

The manuscript could benefit from some careful proofreading, e.g. for grammatical errors (line 369: "have a special structure").

References:

Chen RT, Rubanova Y, Bettencourt J, Duvenaud DK. Neural ordinary differential equations. Advances in neural information processing systems. 2018;31.

Rackauckas C, Ma Y, Martensen J, Warner C, Zubov K, Supekar R, Skinner D, Ramadhan A, Edelman A. Universal differential equations for scientific machine learning. arXiv preprint arXiv:2001.04385. 2020 Jan 13.

Ghadami A, Epureanu BI. Data-driven prediction in dynamical systems: recent developments. Philosophical Transactions of the Royal Society A. 2022 Aug 8;380(2229):20210213.

Greydanus S, Dzamba M, Yosinski J. Hamiltonian neural networks. Advances in neural information processing systems. 2019;32.

McGreivy N, Hakim A. Invariant preservation in machine learned PDE solvers via error correction. arXiv preprint arXiv:2303.16110. 2023 Mar 28.

Cranmer K, Brehmer J, Louppe G. The frontier of simulation-based inference. Proceedings of the National Academy of Sciences. 2020 Dec 1;117(48):30055-62.

**Questions:**

Are step sizes treated as fixed on the backward pass, or do we somehow take gradients w.r.t step size calculations? If the former, how large are the resulting errors compared to the errors in backward propagated states as computed with a non-time-reversible integrator of the same order?

Line 505: "The potential is obtained by integrating the neural network numerically" -- please explain.

The description of Heun's method (line 213) is both very terse (what is it?) and very detailed (initilaization values etc.)

---

> ### Author Response · Authors · 2024-11-27
>
> Dear reviewer, we would like to thank you for the helpful comments and to address your concerns.
> 1. **Paper doesn't provide significant theoretical results, the experimental results are weak.**  We have developed the first, in its kind, class of higher order time reversible integration methods allowing adaptive time stepping. The adaptive stepping in turn permits to considerably accelerate computations. Notice that such methods in combination with the neural ODE approach are especially important in learning models of high dimensional systems exhibiting complex behaviour, such as molecular systems. Using checkpointing would lead to memory explosion in this case, whereas higher order reversible methods would allow to solve the learning problem with low computational costs. With the numerical examples we show that higher order reversible method outperform the known lower order method in terms of the computational time. Notice that the nonlinear oscillations example can be seen as a simplified model of molecular dynamics.
> 2. **Literature review and citations are inadequate.** We have extended the literature review.
> 3. **Never compared to alternative methods such as a non-reversible integrator** We have included in Appendix C the comparison with the Runge-Kutta 4(5) (RK45) method. On the Kepler example (Fig2 and Fig3), Y4 outperforms RK45 . On the other examples (Fig5 and Fig9) RK45 is faster but has a higher memory cost. This highlights the applicability domain of the proposed class of compositional methods. They permit to make computations faster, when the higher order of the integration is required and at the same time they have constant memory costs, which is not possible with the standard methods such as RK45.
> 4. **For parameter estimation tasks, consider accuracy with regard to the reference parameters** We provided additional results showing that the higher order method converges to parameters of high order of accuracy faster than the lower order method. This is displayed in Fig2, Fig3, Fig5
> 5. **A fixed step size was used in table 2** Table 2 shows the results for the adaptive time stepping.
> 6. **Fig2 exaggerates the benefits of Y4 over ALF** The figure was moved to the appendix. It illustrates the need for smaller time-steps in the integration for highly accurate parameter estimation. Additional discussion and clarification was added and can now be found in the Appendix C1
> 7. **The problems are fairly easy.** We have included a new example of discretized PDE. The obtained results displayed in Table 4 show that the higher order method is faster in getting to the predefined training accuracy. In additional, Fig9-10 support the observation.
> 8. **Questions on computation of $d\phi_{j+1}/d\phi_{j}$ in backprop** In Appendix B we have included an alternative approach for the integration of the adjoint, which does not require computation of $\frac{\partial \phi_{k+1}}{ \partial \phi_k}$ for the backpropagation. The method is based on the composition of rescaled steps backward of ALF method. As a result, computation of $\frac{\partial \phi_{k+1}}{ \partial \phi_k}$ is replaced by the  composition of the terms depending of $\frac{\partial f(z,t)}{ \partial z}$. Notice that the Jacobian of $f$ is sparse in the case of discretized PDEs and the memory can be allocated more efficiently.
> 9. **Checkpointing.** We have included a paragraph on the checkpoitning in the end of the description of the gradient computations in the end of Section 2.2. In this paragraph we explain the difference between the approach by checkpointing and by the reversible methods. Checkpointing is highly efficient in combination with high order methods for the tasks, which require high accuracy in learning and use short trajectories in training. However, in case of systems with complex behaviour it is important to consider long time integration which leads to the linear growth in memory with respect to the time of integration. This is the context in which the reversible methods are required.
> 10. **Simulation-based inference.** We have added a discussion on statistical inference to Section 4.1 and related simulation-based inference to the proposed method.
> 11. **The notation in 2.1 is confused.** The section was shortened and the notations were improved.
> 12. **Proofreading** We have corrected mistakes.
> 13.  **Are step sizes treated as fixed on the backward pass?** The time steps are computed in the propagation forward and saved in the memory. In the propagation backward the steps are treated as fixed, see the paragraph on adaptive stepping in Section 3 and Algorithm 2.
> 14. **Line 505** The neural network approximates the gradient of the potential, so to obtain the potential we need to integrate the function given by the neural network and we need to do it numerically, that is, to use a numerical integration methods.
> 15. **Heun's method** We removed the sentence on the initialization and only left the description of the important properties.

---

### Meta-Review · Area_Chair_8eLK · 2024-12-19

**Metareview:**

The paper introduces a higher order reversible time integrator for computing the gradients in the neural ODE setup to train neural networks.   They show results demonstrating the computational savings for the higher order integration with adaptive time stepping compared to ALF steps (prior work for lower order time integration).
The main strengths of this paper are in the positive results for higher order integration that can have significant impact for complex dynamical systems and good explanation of methods that allows for easy understanding from the reader.
The main weakness of this paper is limited contributions - though very promising, reviewers agreed that experimental evidence falls short due to the focus on simplistic systems.

**Additional Comments On Reviewer Discussion:**

Reviewers primarily raised the issue of limited evaluation on simplistic systems. Since the main contribution is in higher order integration (known from numerical methods/analysis), the paper could be made stronger on evaluating on more complex systems. Reviewer 6eWA also raised concerns in employing reversible integrators for learning irreversible dynamics. Since the paper mainly extends prior work on lower order integrators, the authors did not focus on responding to this since this issue is inherent in any reversible integrator. However,  6eWA's concerns are justified and related to the first concern on limited evaluation - I think choosing more complex systems would have helped the authors answer this as well. Given the limited time for rebuttals, this is a challenging request but one the authors can include in future versions of their paper to make their contributions strong.

Reviewers also raised concerns on limited literature review and specific technical questions that the authors have addressed in their rebuttal.

---

### Decision · Program_Chairs · 2025-01-22

Reject